# Sampling from multi-modal distributions with polynomial query complexity in fixed dimension via reverse diffusion

**Adrien Vacher**
CREST, ENSAE
Institut Polytechnique de Paris
adrien.vacher@ensae.fr

**Omar Chehab**
CREST, ENSAE
Institut Polytechnique de Paris
omar.chehab@ensae.fr

**Anna Korba**
CREST, ENSAE
Institut Polytechnique de Paris
anna.korba@ensae.fr

## Abstract

Even in low dimensions, sampling from multi-modal distributions is challenging. We provide the first sampling algorithm for a broad class of distributions — including all Gaussian mixtures — with a query complexity that is polynomial in the parameters governing multi-modality, assuming fixed dimension. Our sampling algorithm simulates a time-reversed diffusion process, using a self-normalized Monte Carlo estimator of the intermediate score functions. Unlike previous works, it avoids metastability, requires no prior knowledge of the mode locations, and relaxes the well-known log-smoothness assumption which excluded general Gaussian mixtures so far.

## 1   Introduction

Sampling from a distribution whose density is only known up to a normalization constant is a fundamental problem in statistics. Formally, given some potential $V : \mathbb{R}^d \to \mathbb{R}$ such that $\int e^{-V(x)}\mathrm{d}x < \infty$, the sampling problem consists in obtaining a sample from some distribution $p$ such that $p$ is $\epsilon$-close to the target $\mu \propto e^{-V}$ with respect to some divergence while maintaining the complexity, i.e., the number of queries to $V$ and possibly to its derivatives, as low as possible. Depending on the shape of the distribution, the typical complexity of existing sampling algorithms can significantly differ.

**Log-concave and "PL-like" distributions**   As in Euclidean optimization, a common assumption in the sampling literature is to assume that $\mu$ is log-concave and log-smooth or equivalently, that $V$ is convex and smooth. Specifically, when the potential $V$ is assumed to be $\alpha$-strongly convex and to have an $L$-Lipschitz gradient, the popular Unajusted Langevin Algorithm (ULA) is known to achieve fast convergence [Durmus and Moulines, 2017, Dalalyan and Karagulyan, 2019]. Because strong-log concavity implies uni-modality, thus excluding many distributions of interest, this assumption was further relaxed to $\mu$ verifying an $\alpha^{-1}$-log-Sobolev inequality, later interpreted as a Polyak–Łojasiewicz type condition on $\mathrm{KL}(\cdot|\mu)$ for the Wasserstein geometry [Blanchet and Bolte, 2018]. Under these conditions, ULA was shown to achieve $\epsilon$-error in Kullback-Leibler (KL) divergence in $\tilde{O}(L^2\alpha^{-2}d\epsilon^{-1})$ queries to $\nabla V$ [Vempala and Wibisono, 2019]. While these polynomial guarantees

39th Conference on Neural Information Processing Systems (NeurIPS 2025).

do go beyond the uni-modal setting, we show in the next paragraph that most existing algorithms still fail to sample from truly multi-modal distributions.

**Multi-modal distributions**   Designing sampling algorithms for multi-modal distributions is an active area of research. However, most existing sampling algorithms are limited in at least two of the following ways:

1. The query complexity is exponential in the parameters of the problem. For instance, when the global $\alpha$-strong convexity of $V$ is relaxed with $\alpha$-strong convexity outside a ball of radius $R$, the log-Sobolev constant of $\mu$ degrades to $O(e^{8LR^2}\alpha)$ [Ma et al., 2019, Prop. 2]. In practice, this does not simply translate to poor worst case bounds: practitioners are well aware that when dealing with multi-modal distributions, ULA-based algorithms suffer from *metastability*, where they get stuck in local modes, leading to slow convergence [Deng et al., 2020].

2. Guarantees are obtained under a log-smoothness assumption; we show in Sec. 3.2 that even for Gaussian Mixtures, which is arguably the most basic multi-modal model, this assumption may not be verified.

3. Explicit *a priori* knowledge on the target distribution is required for the algorithm to converge. For instance, importance sampling requires a proposal distribution whose "effective" support must cover sufficiently well the one of the target; however, since $V$ can only be evaluated point-wise, the access to such a support is unclear. In fact, this limitation goes beyond the multi-modal setting in some cases: no matter the log-Sobolev constant of the target, similarly to euclidean gradient descent, ULA still requires an *a priori* bound on the inverse smoothness constant of $V$ and sufficiently small step size in comparison in order to converge.

Generally speaking, finding a sampling algorithm that addresses even the first limitation is not possible. Recent results provide lower-bounds on the complexity of sampling from a multi-modal distribution: they are exponential in the dimension, as shown in Lee et al. [2018, Th K.1], Chak [2024, Th 3.3] and He and Zhang [2025, Th. 3]. However, when the dimensionality is fixed, the question of whether the three limitations can be addressed remains. *In fixed dimension, can we sample from a broad class of multi-modal distributions with a polynomial number of queries in the parameters of the problem, (e.g. $L, R, \alpha$) and without prior knowledge on these parameters*?

**Our contributions**   Our work answers this question positively. We provide a sampling algorithm that addresses the three limitations outlined above. First, we show that our algorithm has *polynomial* complexity in all the problem parameters but the dimension, thus enabling to efficiently sample from highly multi-modal distributions in fixed dimension. Second, we show that unlike most existing results, our guarantees hold under relaxed regularity assumptions that cover *general* Gaussian mixtures. Third, our algorithm yield guarantees *without* prior knowledge on the parameters of the distribution.

The workhorse of our algorithm is made of two key ingredients: the reverse diffusion scheme, that transfers the sampling problem into a score estimation problem with different levels of noise, and self-normalized importance sampling, to estimate these noisy scores using only query access to $V$ and additional Gaussian samples. Both reverse diffusion [Huang et al., 2024a,b, He et al., 2024] and self-normalized importance sampling [Huang et al., 2025, Jiao et al., 2021, Ruzayqat et al., 2023, Ding et al., 2023, Saremi et al., 2024] were already used separately to sample in a Bayesian setting, where only the log-density of the target is available. However, these works either offer no theoretical guarantees or fail to improve upon existing results. In this article, we combine both reverse diffusion and self-normalized importance sampling in a single algorithm, allowing us to recover the first polynomial time guarantees to sample from highly multi-modal distributions in fixed dimension.

This paper is structured as follows. First, we survey related work on multi-modal sampling in Section 2 and present our main result in Section 3. Then, we detail our sampling algorithm in Section 4 and provide the key ingredients for the proof in Section 5.

## 2 Related work

Before detailing our sampling algorithm in Section 4, we review the main alternatives to ULA when sampling from a multi-modal distribution and the guarantees they offer. We show that existing approaches suffer from at least two of three drawbacks outlined in the introduction: exponential query complexity, a restrictive smoothness assumption that excludes general Gaussian mixtures (as shown later in Sec. 3.2), and unavailable prior knowledge of the distribution.

**Proposal-based algorithms** At a high level, proposal-based algorithms use a proposal distribution that is easy to sample from and that can either be directly used as a proxy for the target, or, alternatively, whose samples will be rejected (respectively re-weighted) to obtain approximate samples from the target; the corresponding algorithm for the latter case is the well-known rejection sampling (respectively importance sampling) scheme. Guarantees for these methods typically assume that the target distribution is log-smooth. Furthermore, they must use carefully designed proposals that require prior knowledge of the target distribution that is often unavailable in practice.

For instance, when the target is $L$-log-smooth and with finite second moment $m_2$, one can design and sample from a proxy distribution that is $\epsilon$-close in TV to the target using $\tilde{O}((Lm_2\epsilon^{-1})^{O(d)})$ queries to the potential function $V$ and its gradient $\nabla V$ [He and Zhang, 2025]. While this bound is indeed polynomial in fixed dimension, the design of the proxy requires an $\epsilon$-approximation of the global minimum of $V$ which can only be achieved if $m_2$, therefore the location of the mass, is explicitly available; this is rarely the case in practice.

Similarly, if we further assume that the target is $\alpha$-strongly log-concave outside of a ball of radius $R$, one can achieve an $\epsilon$-precise approximation of the target distribution in TV with polynomial number of queries to $\nabla V$ when $d$ is fixed via an importance sampling scheme [Chak, 2024, Th. 2.3]. In this case, the proposal is such that it coincides with the target when the potential of the target is above some cutoff, and is flat elsewhere. While there exists a cutoff value ensuring the log-concavity of the proposal, thus allowing its efficient sampling, this value depends on the unknown constants $L, R$ and $\alpha$ [Chak, 2024, Prop. 5.1].

**Tempering-based algorithms** Instead of directly sampling from $\mu \propto e^{-V}$, these algorithms start by sampling from a flattened version of $\mu$ given by $\mu^\beta \propto e^{-\beta V}$ for small $\beta$, and gradually increase $\beta$ to 1, a strategy sometimes also referred to as annealing. When $\mu$ is assumed to be a finite mixture of the same shifted $\alpha$-strongly log-concave and $L$-log-smooth distribution, Lee et al. [2018] proved that for a well-chosen (stochastic) sequence of flattened distributions $\mu^{\beta_i}$, sampling up to precision $\epsilon$ in KL can be achieved in $poly(L, \alpha^{-1}, w_{min}^{-1}, d, \epsilon^{-1}, R)$ queries to $\nabla V$, where $R$ is the location of the furthest mode and $w_{min}$ the minimum weight in the mixture. However, this setting is quite restrictive: for Gaussian mixtures for instance, it only handles the case where the covariance matrices are identical for all components. Furthermore, the algorithm requires explicit knowledge of $R$ which is unavailable.

**Föllmer Flows** Instead of directly sampling from the target distribution $\mu \propto e^{-V}$, Föllmer Flows start from a simple (e.g. Gaussian or Dirac mass) distribution that is progressively interpolated to the target using a Schrödinger bridge. When the initial distribution is a Dirac mass at 0, this bridge solves a closed-from SDE [Wang et al., 2021, Theorem 3] which can thus be discretized to generate samples from the target. In the context of Bayesian inference, where one only has access to the unnormalized density, Vargas et al. [2022] estimated the drift of the resulting SDE via neural methods; in particular no guarantees on the sampling quality are provided. In Huang et al. [2025], Jiao et al. [2021], Ruzayqat et al. [2023], Ding et al. [2023], the drift is estimated via a Monte-Carlo method. While in appearance, these works provide strong polynomial guarantees, a closer look shows that these guarantees only hold if the function $f(x) = e^{-V(x)+\|x\|^2/2}$ is Lipschitz, smooth and bounded from below; we show in Appendix B.1.2 that this assumption is quite restrictive.

**Diffusion-based methods** Over the past few years, diffusion-based algorithms, and especially reverse diffusion, that we shall review in details in Sec. 3, have emerged as solid candidates for multi-modal sampling. In essence, they allow to transfer the sampling problem into the problem of estimating the scores of intermediate distributions that are given by the convolution of the initial distribution with increasing levels of Gaussian noise. Under an $\epsilon$-oracle of these intermediate scores,

it has been shown that diffusion-based methods could yield an $\epsilon$-approximate sample of the target in $\text{poly}(\epsilon^{-1})$ time under milder and milder assumptions [Chen et al., 2023b,a, Benton et al., 2024, Li et al., 2024, Conforti et al., 2025, Gentiloni-Silveri and Ocello, 2025, Cordero-Encinar et al., 2025]. This framework has been applied with tremendous empirical success in generative modeling, where numerous samples of the target are already available and one seeks to produce new samples from the target, for several years now [Song and Ermon, 2019, Ho et al., 2020, Bortoli et al., 2021, Chen et al., 2024]. However, it was only quite recently that this framework has been applied in a Bayesian context, where only an unnormalized density of the target, instead of samples, is available [Huang et al., 2024a,b, He et al., 2024, Grenioux et al., 2024, Akhound-Sadegh et al., 2024]. In a work closely related to ours, Huang et al. [2024b] showed that if the intermediate scores remain $L$-log-smooth for any noise level, which in particular implies that the target itself is $L$-log-smooth, their algorithm could reach a complexity of $O(e^{L^3 \log^3((Ld+m_2)/\epsilon)})$ with $m_2$ the second order moment of the target. In He et al. [2024], the smoothness assumption is relaxed with a sub-quadratic growth assumption $V(x) - V(x^*) \leq L\|x - x^*\|^2$, where $x^*$ is any global minimizer of $V$, yet at the price of an oracle access to $V(x^*)$; under these assumptions, the authors manage to obtain a complexity that is at best $O(L^{d/2}\epsilon^{-d}e^{L\|x^*\|^2 + \|x_N\|^2})$ where $x_N$ is the final output sample. Under the reasonable (and desirable) assumption that $\mathbb{E}[\|x_N\|^2] \approx m_2$, Jensen's inequality yields an overall $O(L^{d/2}\epsilon^{-d}e^{L\|x^*\|^2 + m_2})$ complexity. In particular, both these works suffer at least two of the limitations mentioned in the introduction, making them ill-suited for multi-modal sampling.

## 3 Presentation of the main result and application to Gaussian Mixtures

### 3.1 Main result

As in the works mentioned above, we rely on the recent advances on reverse diffusion and focus on the task of estimating the intermediate scores. Using an estimator that is described in Sec. 4, we recover polynomial sampling guarantees for densities verifying the assumptions described hereafter.

**Assumption 1** (Semi-log-convexity) *We assume that $\mu \propto e^{-V}$ is such that $\log(\mu)$ is $C^2$ and verifies $\nabla^2 \log(\mu) \succeq -\beta I_d$ or equivalently $\nabla^2 V \preceq \beta I_d$ for some $\beta \geq 0$.*

This assumption shall be referred to as *semi-log-convexity*, by analogy with the functional analysis literature [Mikulincer and Shenfeld, 2023, Theorem 3]. Note that is has also been referred to as 1-sided Lipschitzness for $\nabla V$ [Gentiloni-Silveri and Ocello, 2025]. It relaxes the classical log-smoothness assumption which implies the additional lower bound $\nabla^2 V \succeq -\beta I_d$. In particular, unlike the latter, a mixture of semi-log-convex densities remains semi-log-convex [Marshall et al., 1979, Chap. 16.B]; we provide a quantitative version of this statement in Sec. 3.2 in the case of Gaussian mixtures.

**Assumption 2** (Dissipativity) *We assume that $\mu \propto e^{-V}$ is such that there exists $a > 0, b \geq 0$ for which its potential satisfies $\langle \nabla V(x), x \rangle \geq a\|x\|^2 - b$ for all $x \in \mathbb{R}^d$.*

This assumption is referred to as *dissipativity* as common in the sampling and optimization literature [Raginsky et al., 2017, Zhang et al., 2017, Erdogdu and Hosseinzadeh, 2021]. Note that this assumption relaxes strong convexity outside of a ball which can be equivalently re-written as $\langle \nabla V(x) - \nabla V(y), x - y \rangle \geq a\|x - y\|^2 - b$ **for all pairs** $(x, y) \in \mathbb{R}^d$, also referred to as *strong dissipativity* [Eberle, 2013, Erdogdu et al., 2022]. We show in Sec. 3.2 that unlike strong-dissipativity, a mixture of dissipative distributions remain dissipative.

**Theorem 1** *[Main result, informal] Suppose that Assumption 1 and 2 hold. Then, for all $\epsilon > 0$, there exists a stochastic algorithm whose parameters only depend on $\epsilon$ (and not on the parameters of the problem), that outputs a sample $X \sim \hat{p}$ such that $\mathbb{E}[\text{KL}(\mu, \hat{p})] \lesssim \epsilon\beta^{d+3}(b + d)/a^2$ in $O(\text{poly}(\epsilon^{-d}))$ queries to $V$, where $\lesssim$ hides a universal constant as well as log quantities in $d, \epsilon^{-1}, \beta, a, b$.*

*In particular, when $d$ is fixed, this algorithm can output a sample from a distribution that is $\epsilon$-close to $\mu$ in expected* KL *in $\text{poly}((b + 1)/a, \beta, \epsilon^{-1})$ running time.*

Our algorithm addresses the three limitations outlined in the sections above. When the dimension $d$ is fixed, we obtain a *polynomial* query complexity. This guarantee does *not* assume log-smoothness and applies to general Gaussian mixtures, as will be shown in the next subsection. Moreover, this guarantee does *not* require running the algorithm with any explicit knowledge of the target distribution's constants $a, b, \beta$.

| Algorithm | Assumptions | Oracle | Complexity (Total Variation) |
|---|---|---|---|
| ULA [Ma et al., 2019] | $((\alpha+L)\mathbf{1}_{\|x\|>R} - L)I_d \preceq \nabla^2 V(x) \preceq LI_d$ | $\nabla V, L$ | $O(e^{16LR^2}(L/\alpha)^2 d\epsilon^{-1})$ |
| Proposal-based [He and Zhang, 2025] | $\|\nabla^2 V\| \le L, m_2 \le M$ | $V, \nabla V, M, L$ | $O((LM\epsilon^{-1})^{O(d)})$ |
| RD + ULA [Huang et al., 2024b] | $\|\nabla^2 \log(p_t)\| \le L, m_2 < \infty$ | $\nabla V$ | $e^{O(L^3 \log^3((Ld+m_2/\epsilon)))}$ |
| RD + Rejection Sampling [He et al., 2024] | $V(x) - V(x^*) \le L\|x - x^*\|^2, m_2 < \infty$ | $V, V^*$ | $O(L^{d/4}\epsilon^{-d/2}e^{(L\|x^*\|^2+m_2)/2})$ |
| RD + Self-normalized IS (ours) | $\nabla^2 V \preceq \beta I_d, \langle \nabla V(x), x\rangle \ge a\|x\|^2 - b$ | $V$ | $O(\sqrt{d}\beta^{(d+3)/2}\sqrt{b+d}/a\epsilon^{-(d+2)})$ |

Table 1: **Complexity of sampling algorithms.** We denote by $x^*$ a global minimizer of $V$, by $V^* = V(x^*)$ the global minimum of $V$, by $m_2$ the second moment of the target, by $p_t$ the density of the forward process (see Sec. 4), and $\|\cdot\|$ the operator norm for matrices. In RD + ULA, RD refers to a Reverse Diffusion algorithm and ULA to how the intermediate scores are estimated. Even though originally stated in KL for our work and the one of He et al. [2024], all the complexities are w.r.t. the Total Variation distance (obtained via the Pinsker inequality).

**Overall comparison**   We summarize in Table 1 how our algorithm compares to previous approaches in terms of assumptions, oracles required to run the algorithm and resulting complexity. Along with the work of He and Zhang [2025], our algorithm is the only one that is polynomial in the parameters of the distribution when $d$ is fixed. Furthermore, while our dissipativity assumption is stronger than finite second moment, we relax the log-smoothness assumption by semi-log-convexity which notably covers general Gaussian mixtures. Finally, as mentioned above, because their algorithm requires an $\epsilon$-approximation of the global minimum of $V$, they require an explicit upper-bound on the second order moment of $\mu$ which may not be available in practice.

**Numerical illustration**   In Figure 2, we consider a standard task in the literature: sampling from a mixture of 16 equally weighted Gaussians with unit variance and centers uniformly distributed in $[-40, 40]^2$ [Midgley et al., 2023]. We compare our algorithm against Unadjusted Langevin Algorithm (ULA) and to the reverse diffusion algorithm of Huang et al. [2024a] (RDMC). We also implemented the zeroth-order method of He et al. [2024], but it failed to converge. ULA was initialized from $\mathcal{N}(0, I_2)$ and run for $5 \times 10^4$ steps. All three methods used the same discretization step size $h = 0.01$. Our reverse diffusion algorithm and RDMC were both run with 500 reverse diffusion steps and both used 100 samples to estimate the intermediate scores. As discussed in Sec. 4, while our samples are simply drawn from a Gaussian distribution, the samples used in RDMC are drawn from a auxiliary, multi-modal distribution generated via an inner ULA step. In this experiment, this inner ULA was initialzed with a standard Gaussian and was run for 100 steps; in particular, we emphasize that while ULA and our algorithm were roughly given the same computational budget, the one given to RDMC was a hundred times superior. As a result, ULA and our algorithm took approximately one minute to run on a computer locally and the RDMC method required over an hour. We observe that unlike the two others, our algorithm successfully recovers all the modes.

In Figure 1, we monitor the convergence of the same three algorithms as a function of the problem difficulty, measured by the distance between the modes. Here, the target distribution is a mixture of three Gaussians in two dimensions. It has equal weights $1/3$, equidistant modes located at a distance $R$ from the origin, and different covariances $(I, I/2, I/4)$. The final error is measured in Wasserstein distance, because it can be easily approximated using samples: we use 500 samples generated by the sampling algorithm and 500 samples from the true target distribution. We allowed each of the three algorithms $10^6$ queries to the potential $V$ or to its gradient:

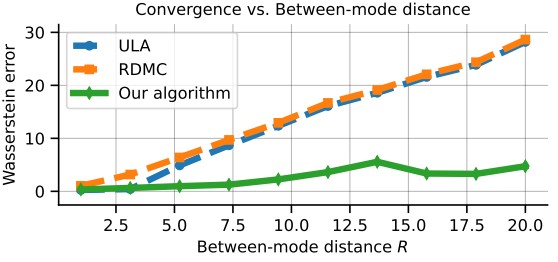

Figure 1: Error in Wasserstein distance as a function of the between-mode distance.

we performed $10^6$ iterations for ULA, we used 100 steps of reverse diffusion for our algorithm and RDMC. We used 100 inner steps of ULA for RDMC that was initialized with a standard Gaussian. We used 100 particles for score estimation for RDMC and 10000 for our algorithm. As expected, as the between-mode distance grows with $R$, our algorithm yields the lowest error.

## 3.2 Application to Gaussian mixtures

We now apply our results to derive provable sampling guarantees for general Gaussian mixtures. Apart from the very recent exception of Lytras and Mertikopoulos [2025] that we discuss below, note that despite their wide popularity, no sampling guarantees were yet derived for general Gaussian mixtures. Indeed, unless they verify some specific assumptions such as identical covariance matrices among components [Cordero-Encinar et al., 2025, Lemma B.1], general Gaussian mixtures do not satisfy the log-smoothness assumption, thus they do not fit the framework of many previously discussed works. In fact, their gradient may not even be Hölder continuous: consider the simple counter-example of a two-dimensional mixture $\mu = 0.5\mathcal{N}(0, \Sigma_1) + 0.5\mathcal{N}(0, \Sigma_2)$ with covariances $\Sigma_1 = diag(1, 0.5)$ and $\Sigma_2 = diag(0.5, 1)$. On the diagonal $x = y$, the score is $\nabla \log(\mu)(x, x) = -3/2(x, x)$, while near the diagonal, right above it for instance, the score behaves asymptotically as $\nabla \log(\mu)(x, x) \sim_{x \to +\infty} -(2x, x)$; we provide a rigorous analysis in Appendix B.2. Fortunately though, Gaussian mixtures do verify Assumptions 1-2, as we next show.

**Proposition 2** *Let $\mu = \sum_{i=1}^{p} w_i \mathcal{N}(\mu_i, \Sigma_i)$ and denote $\lambda_{min} > 0$ (resp. $\lambda_{max}$) the minimum (resp. maximum) eigenvalue of the covariance matrices $\Sigma_i$. It holds that $-\nabla^2 \log(\mu) \preceq I_d/\lambda_{min}$ and that for all $x \in \mathbb{R}^d$, $\langle -\nabla \log(\mu)(x), x \rangle \geq \|x\|^2/(2\lambda_{max}) - \lambda_{max} \max_i(\|\mu_i\|/\lambda_{min})^2$.*

The proof is deferred to Appendix B.3. Combined with Theorem 1, Proposition 2 shows that we can sample any Gaussian mixture with average precision $\epsilon$ in KL in $O(\text{poly}(\kappa, R, d, \lambda_{min}^{-d}, \epsilon^{-d}))$ queries to $V$ where $R = \max_i \|\mu_i\|$ and $\kappa = \lambda_{max}/\lambda_{min}$. There exists a relatively recent literature seeking to relax the log-smoothness assumption. For instance, Chatterji et al. [2019], Erdogdu and Hosseinzadeh [2021], Nguyen et al. [2021] work under weak smoothness, i.e. $\alpha$-Hölder continuous gradient of the potential for $\alpha$ in $[0, 1]$ (recall that $\alpha$=1 recovers the smooth case). However, as shown above, this relaxation is not sufficient yet to cover general Gaussian mixtures. The only reference we know of that does is the work of Lytras and Mertikopoulos [2025], who used a regularized version of Langevin to relax the global smoothness condition with local Lipschitz smoothness and polynomial growth of the Lipschitz constant. Yet crucially, their complexity bound scales as a polynomial of the *Poincaré* constant of $\mu$. We prove in Appendix B.4 that for the mixture $\mu = 0.5\mathcal{N}(Ru, \lambda_{\max}I_d) + 0.5\mathcal{N}(-Ru, \lambda_{\min}I_d)$ with $u$ any unit vector and $0 < \lambda_{\min} \leq \lambda_{\max}$, this constant is at least $(R^2 e^{R^2/(2\lambda_{\max})})/2$. In particular, this method still degrades exponentially with the multi-modality parameter $R$.

## 4 Our sampling algorithm

In this section, we introduce the reverse diffusion framework and explain how it reduces the sampling problem to that of estimating the scores along the forward Ornstein-Uhlenbeck (OU) process.

### 4.1 Reverse diffusion: from sampling to score estimation

Reverse diffusion methods emerged as an alternative to Langevin-based samplers in order to overcome metastability and were first introduced to the ML community in Song et al. [2021]. They rely on the so-called forward process

$$\begin{cases} \mathrm{d}X_t = -X_t\mathrm{d}t + \sqrt{2}\mathrm{d}B_t \\ X_0 \sim \mu, \end{cases} \tag{1}$$

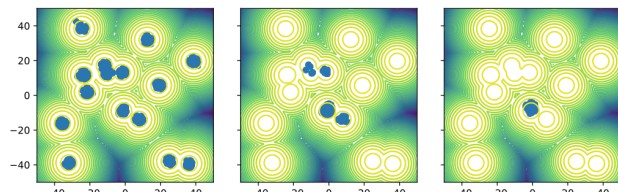

Figure 2: From left to right: our algorithm vs. ULA vs. Huang et al. [2024a]. The color scheme indicates the probability density value of the distribution we want to sample from (dark is low probability density, bright is high probability). The blue dots are the samples produced by the algorithm.

which corresponds to the standard OU process initialized at $\mu$, that is a specific case of a Langevin diffusion targeting a standard Gaussian, that we will denote $\pi$. Note that since the target of this process, the standard Gaussian, is 1-strongly log-concave, the resulting process converges exponentially fast to the equilibrium. In order to sample from $\mu$, reverse diffusion algorithms rely on the semi-discretized *backward process*: given a horizon $T$ that we discretize as $0 = t_0 \leq t_1 \leq \cdots t_{N-1} \leq t_N = T$, the latter writes

$$\mathrm{d}Y_t = Y_t \mathrm{d}t + 2\nabla \log(p_{t_{N-k}})(Y_t)\mathrm{d}t + \sqrt{2}\mathrm{d}B_t \,, \ \ t \in ]T - t_{N-k}, T - t_{N-(k+1)}] \,, \tag{2}$$

with $Y_0 \sim p_T$ and where $p_t$ is the distribution of the forward process Eq. 1 at time $t$. Note that this reverse process cannot be readily implemented for two reasons: first, it requires the knowledge of the intermediate scores $\nabla \log(p_{t_k})$ which are not available in closed form. Second, it requires sampling from the distribution $p_T$. Nevertheless, if one can access a proxy $s_{t_k}$ of the scores $\nabla \log(p_{t_k})$, and considering $T$ large enough so that $p_T \approx \pi$, we can implement instead

$$\mathrm{d}Y_t = Y_t \mathrm{d}t + 2s_{t_k}(Y_k)\mathrm{d}t + \sqrt{2}\mathrm{d}B_t \,, \ \ t \in ]T - t_{N-k}, T - t_{N-(k+1)}] \,, \tag{3}$$

with $Y_0 \sim \pi$ and where all iterations can be solved in closed form. Because the forward process Eq. 1 converges exponentially fast, we can expect the initialization error $Y_0 \sim \pi$ instead of $Y_0 \sim p_T$ to be small after a short time $T$. Furthermore, if the proxies $s_{t_k}$ are sufficiently accurate, one can expect that the process output by the approximate scheme Eq. 3 has a distribution that is close to the target $\mu$. Over the past three years, several works provided quantitative bounds of the error induced by the discretization, the use of an approximate score and the initialization error with respect to different divergences and under various assumptions [Bortoli et al., 2021, Lee et al., 2022, Chen et al., 2023b,a, Conforti et al., 2025]. Yet, we shall rely exclusively on the following theorem as it is the most suited to our framework.

**Theorem 3** (Conforti et al. [2025]) *Assume that $\mu \propto e^{-V}$ has finite Fisher-information w.r.t. $\pi$ the standard gaussian density in $\mathbb{R}^d$:*

$$\mathcal{I}(\mu, \pi) := \int \|x - \nabla V(x)\|^2 \mathrm{d}\mu(x) < +\infty \,.$$

*Then, for the constant step-size discretization $t_k = kT/N$, denoting $p$ the distribution of the sample $Y_T$ output by Eq. 3, it holds that*

$$\mathrm{KL}(\mu, p) \lesssim (d + m_2)e^{-T} + \frac{1}{N}\sum_{k=1}^{N} \|\nabla \log(p_{t_k}) - s_{t_k}\|_{L^2(p_{t_k})}^2 + \frac{T}{N}\mathcal{I}(\mu, \pi) \,,$$

*where $m_2$ is the second order moment of $\mu$ and where $\lesssim$ hides a universal constant.*

The previous theorem shows that under mild assumptions that notably allow for multi-modality, the problem of sampling from $\mu$ can be transferred into a score approximation problem along the forward process. In the next subsection, we present an estimator for these intermediate scores that is tractable given the knowledge of the unnormalized density $\mu \propto e^{-V}$.

## 4.2 Derivation of an estimator

The key observation to derive an estimator of the intermediate scores is that the forward process Eq. 1 is nearly available in closed form. However, this closed form may be written in different manners.

**Different expressions of the scores** Consider Eq. 1 integrates to $X_t = \sqrt{\lambda_t}X_0 + \sqrt{1 - \lambda_t}Z$, where $\lambda_t = e^{-2t}$, $Z$ is a standard $d$-dimensional Gaussian, and $X_0$ is a random variable simulating the target distribution. The corresponding density convolves the target density with a Gaussian, as

$$p_t^V(x) = \frac{1}{\sqrt{1 - \lambda_t}}\pi\left(\frac{x}{\sqrt{1 - \lambda_t}}\right) * \frac{1}{\sqrt{\lambda_t}}\mu\left(\frac{x}{\sqrt{\lambda_t}}\right) \,, \tag{4}$$

where $\pi$ is a standard Gaussian and $\mu(\cdot) \propto \exp(-V(\cdot))$ is the target distribution. From this expression, we can obtain different formulas for the scores. We retain

1. $\nabla \log p_t^V(x) = \frac{1}{\sqrt{\lambda_t}}\mathbb{E}_{y|x}\left[\nabla \log \mu\left(\frac{x-y}{\sqrt{\lambda_t}}\right)\right], \ \ y|x \sim \propto \mu\left(\frac{x-y}{\sqrt{\lambda_t}}\right) \times \mathcal{N}(y; 0, (1 - \lambda_t)I) \,,$

2. $\nabla \log p_t^V(x) = \frac{1}{\sqrt{\lambda_t}} \frac{\mathbb{E}_y\left[\nabla \log \mu\left(\frac{x-\sqrt{1-\lambda_t}\,y}{\sqrt{\lambda_t}}\right)\mu\left(\frac{x-\sqrt{1-\lambda_t}\,y}{\sqrt{\lambda_t}}\right)\right]}{\mathbb{E}_y\left[\mu\left(\frac{x-\sqrt{1-\lambda_t}\,y}{\sqrt{\lambda_t}}\right)\right]}, \quad y \sim \mathcal{N}(y;0,I) \quad,$

3. $\nabla \log p_t^V(x) = \frac{1}{1-\lambda_t}\left(\mathbb{E}_{y|x}[y] - x\right), \quad y|x \sim \mu\left(\frac{y}{\sqrt{\lambda_t}}\right) \times \mathcal{N}(y;x,(1-\lambda_t)I) \quad,$

4. $\nabla \log p_t^V(x) = \frac{1}{\sqrt{1-\lambda_t}} \frac{\mathbb{E}_y\left[y\mu\left(\frac{x+\sqrt{1-\lambda_t}\,y}{\sqrt{\lambda_t}}\right)\right]}{\mathbb{E}_y\left[\mu\left(\frac{x+\sqrt{1-\lambda_t}\,y}{\sqrt{\lambda_t}}\right)\right]}, \quad y \sim \mathcal{N}(y;0,I).$

Derivations can be found in Appendix A. The second [Akhound-Sadegh et al., 2024], third [Huang et al., 2024a, He et al., 2024, Grenioux et al., 2024], and fourth [Saremi et al., 2024] identities have been used to build Monte Carlo estimators of the score. Hence, one natural way to group these identities is by how difficult it is to draw the samples $y$. The second and fourth identities sample from a Gaussian distribution, whereas the first and third identities sample from an auxiliary multi-modal distribution from generating $y|x$. As $t$ gets closer to zero, this auxiliary distribution resembles the original target, so sampling from it progressively becomes as difficult as sampling from the original target density. This observation explains why, unsurprisingly, Huang et al. [2024a], He et al. [2024] do not improve upon existing results. Another way to group these identities is whether or not they require evaluating the score of the target distribution: identities one and two do and are referred to as TSI estimators, referring to the Target Score Identity (TSI) used to obtain them [Bortoli et al., 2024]; identities three and four do not and are referred to as DSI estimators, referring to a Denoising Score Identity (DSI) used to obtain them [He et al., 2025]. Combinations of TSI and DSI estimators have also been considered [Phillips et al., 2024, He et al., 2025].

**A self-normalized estimator** We now focus on the estimator of the scores that we use. It is a rewrite of the fourth identity and is a *ratio* of expectations under Gaussians

$$\nabla \log\left(p_t^V\right)(z) = \frac{-1}{1-e^{-2t}} \frac{\mathbb{E}[Y_t e^{-V(e^t(z-Y_t))}]}{\mathbb{E}[e^{-V(e^t(z-Y_t))}]} \,,$$

with $Y_t \sim \mathcal{N}(0,(1-e^{-2t})I_d)$ which is easy to simulate. While conventional statistical wisdom may suggest using independent samples to estimate both the numerator and the denominator, we voluntarily choose to correlate them and implement instead

$$\hat{s}_{t,n}(z) := \frac{-1}{1-e^{-2t}} \frac{\sum_{i=1}^n y_i e^{-V(e^t(z-y_i))}}{\sum_{i=1}^n e^{-V(e^t(z-y_i))}} \,, \tag{5}$$

where the $y_i$ are independent Gaussians such that $y_i \sim \mathcal{N}(0,(1-e^{1-2t})I_d)$ ; we refer to this estimator as *self-normalized* as common in the sampling literature [Agapiou et al., 2017]. The key property of self-normalized estimators is that they remain nearly bounded: in our case, it holds uniformly in $z$ that

$$\mathbb{E}[\|\hat{s}_{t,n}(z)\|] \leq \frac{\mathbb{E}[\max_i \|y_i\|]}{1-e^{-2t}} \sim \sqrt{\frac{d\log(n)}{1-e^{-2t}}} \,.$$

This boundedness will allow us to derive a non-asymptotical control on the quadratic error that we present in the next section.

We now explain how this differs from previous work. While this estimator was already considered in [Saremi et al., 2024], the authors did not use it within the reverse diffusion pipeline and more importantly, we are the first work to derive quantitative guarantees for this estimator, which is one of our core contributions. Similarly, an estimator close to this one was considered in the context of Föllmer Flows [Jiao et al., 2021, Ruzayqat et al., 2023, Ding et al., 2023] in order to approximate the shift in the corresponding Schrödinger bridge. We show in Appendix B.1.1 that while this shift is itself close to the intermediate scores that we seek to approximate, the resulting self-normalized estimator is more degenerate. Namely, as discussed above, their guarantees are significantly weaker.

## 5 Sketch of proof of the main result

The proof is decomposed in three steps: (i) we derive a non-asymptotic bound on the quadratic error of the estimator presented in Eq. 5 (ii) we show that under Assumptions 1-2, the integrated error of

this estimator (that appears in the bound of Theorem 3) can be fully controlled by the zeroth and second order moments of the ratio $\Phi_t = p_t^{2V}/p_t^V$, where $p_t^{2V}$ (respectively $p_t^V$) is the density of the forward process defined in Eq. 1 initialized at $\mu^2$ (respectively $\mu$) (iii) we provide a quantitative bound on these moments as well as other relevant quantities and we conclude with Theorem 3.

**Proposition 4** (Non-asymptotic bound on the quadratic error) *For all $z \in \mathbb{R}^d$, $n$ and $t > 0$, denoting $p_t^{2V}$ (respectively $p_t^V$) the density of the forward process defined in Eq. 1 initialized at $\mu^2 \propto e^{-2V}$ (respectively $\mu \propto e^{-V}$), $Z_{2V}$ (respectively $Z_V$) the normalizing constant of $\mu^2$ (respectively $\mu$) and $\pi$ the density of the standard Gaussian, it holds that*

$$\mathbb{E}\left[\|\hat{s}_{t,n}(z) - \nabla\log\left(p_t^V\right)(z)\|^2\right] \leq \frac{32e^2(d + \log(n))}{n(1 - e^{-2t})}\left(1 + \theta_t(z)\frac{p_t^{2V}(z)Z_{2V}e^{td}}{(p_t^V(z)Z_V)^2}\right),$$

*with $\theta_t(z) = \Delta\log\left(p_t^{2V}\right)(z) - \frac{\Delta\log(\pi)(z)}{1 - e^{-2t}} + \|\nabla\log\left(p_t^V\right)(z)\|^2 + \|\nabla\log\left(p_t^{2V}\right)(z)\|^2 + 1$.*

The complete proof is left to Appendix C yet we briefly sketch the main arguments. We split the expectation on the event $A$ where the empirical denominator $\hat{D}$ of Eq. 5 (respectively the empirical numerator $\hat{N}$) is not too small (respectively not too large) with respect to its expectation $D$ (respectively $\|N\|$) and on the complementary $\bar{A}$. Over $A$, we use a second-order Taylor expansion to make the variances of both the numerator and the denominator appear, and compute them explicitly. Conversely, the quadratic error of the estimator remains almost bounded on $\bar{A}$. We use Chebyshev's inequality to upper-bound $\mathbb{P}(\bar{A})$ and make the variances of the numerator and of the denominator appear again, which concludes the proof.

**Remark 5** *Generic bounds on self-normalized estimators were derived in Agapiou et al. [2017, Theorem 2.3]. Yet, we show in Appendix B.5 that if used in our context, they would involve at least an extra $1/(p_t^V)$ factor which can cause the integrated error $\int \mathbb{E}\left[\|\hat{s}_{t,n}(z) - \nabla\log\left(p_t^V\right)(z)\|^2\right]\mathrm{d}p_t^V(z)$ to diverge.*

Then, we need to control the Laplacian of the forward processes as well as their gradient appearing in $\theta_t$ in the former proposition. As mentioned in the previous section, the intermediate scores can be re-written as

$$\nabla\log\left(p_t^V\right)(z) = \frac{z - e^{-t}\mathbb{E}_{q_{t,z}}[Y]}{1 - e^{-2t}},$$

with $q_{t,z}(x) \propto e^{-V(x)}e^{-\frac{\|e^{-t}x - z\|^2}{2(1 - e^{-2t})}}$. Now we note that if $\mu \propto e^{-V}$ is dissipative, so is $q_{t,z}$; in particular we can quantitatively bound its second order moment and *a fortiori*, upper-bound $\|\nabla\log\left(p_t^V\right)(z)\|^2$.

**Proposition 6** (Regularity bounds on the forward process) *Suppose that Assumption 2 holds. Then, for all $z \in \mathbb{R}^d$ and $t > 0$, it holds that*

$$\begin{cases} \|\nabla\log\left(p_t^V\right)(z)\|^2 \leq \frac{\|z\|^2}{(1 - e^{-2t})^2}\left(2 + \frac{e^{-2t}}{a(1 - e^{-2t})}\right) + \frac{2e^{-2t}(2b + d)}{a(1 - e^{-2t})^2}, \\ \Delta\log\left(p_t^V\right)(z) \leq \frac{e^{-2t}}{(1 - e^{-2t})^2}\left(\frac{\|z\|^2}{2a(1 - e^{-2t})} + \frac{2b + d}{a}\right). \end{cases}$$

The complete proof is left in Appendix D.2. This result implies that the $\theta_t$ term defined in Proposition 4 is at most of order of order $\theta_t(z) \sim (1 + \|z\|^2)$ w.r.t. $z$. In particular, the average integrated error $\mathbb{E}[\|\hat{s}_{t,n}(z) - \nabla\log\left(p_t^V\right)\|_{L^2(p_t^V)}^2]$ can be upper-bounded with respect to the zeroth and the second order moments of the ratio $\Phi_t = p_t^{2V}/p_t^V$. In the next proposition, we show that these moments are bounded under the semi-log-convexity assumption. By a slight abuse of notation, we shall denote $m_i(\Phi_t) = \int \|x\|^i \Phi_t(z)\mathrm{d}z$ the $i$-th moment of $\Phi_t$.

**Lemma 7** (Bounds on the moments of the ratio) *Assume that $\mu \propto e^{-V}$ has finite second moment $m_2$ and that Assumption 1 holds. Then,*

$$\begin{cases} m_0(\Phi_t) \leq \frac{(Z_V)^2}{Z_{2V}}e^{td}(\beta(1 - e^{-2t}) + e^{-2t})^d, \\ m_2(\Phi_t) \leq \frac{2e^{t(d+2)}(Z_V)^2}{Z_{2V}}(\beta(1 - e^{-2t}) + e^{-2t})^{d+2}\left[m_2 + d(\beta + 1) + 2d\log\left(\frac{\beta\mu(0)^{-2/d}}{2\pi}\right)\right], \end{cases}$$

*where $Z_{2V}$ (respectively $Z_V$) is the normalization constant of $e^{-2V}$ (respectively $e^{-V}$) and where $\pi$ in the right-hand-side refers to the constant $\pi \approx 3.14$.*

The proof of this lemma is deferred to Appendix E. It relies on a key result in Mikulincer and Shenfeld [2023, Lemma 5] where it is shown that for $\beta$-semi-log convex distributions, the following bound holds:

$$\nabla^2 \log(\pi) - \nabla^2 \log\big(p_t^V\big) \preceq \frac{(\beta-1)e^{-2t}}{(1-e^{-2t})(\beta-1)+1} I_d \,.$$

In particular, the right-hand-side remains bounded w.r.t. $\beta$ whenever $t > 0$, which allows to avoid an exponential dependence in $\beta$ in our final bounds. It suffices now to control $\mathcal{I}(\mu, \pi), m_2$ and $\mu(0)$ in order to apply Theorem 3 and conclude.

**Lemma 8** *Assume that* $\mu \propto e^{-V}$ *is such that Assumption 1 and 2 hold:* $\nabla^2 V \preceq \beta I_d$ *and* $\langle \nabla V(x), x \rangle \geq a\|x\|^2 - b$ *for some* $a > 0, b, \beta \geq 0$. *Then,*

$$\begin{cases} m_2 \leq (b+2d)/a \,, \\ \mathcal{I}(\mu, \pi) \leq 2(b+2d)/a + 2\beta d \,, \\ \log\big(\mu(0)^{-2/d}\big) \leq 4\beta b/(da) + 2\pi + \log(2/a) \,. \end{cases}$$

*Here again, the* $\pi$ *in the right-hand-side refers to the constant* $\pi \approx 3.14$.

The proof is left to Appendix D.1. We can now state our main result.

**Theorem 9** *Under Assumptions 1 and 2, if we run the algorithm of Eq. 3 with* $T = \log(1/\epsilon)$, $N = 1/\epsilon$, $t_k = kT/N$ *and with the stochastic score estimators* $\hat{s}_{n_k,t_k}$ *defined in Eq. 5 with* $n_k = d\epsilon^{-(2d+3)}$, *then, denoting* $\hat{p}$ *the stochastic distribution of the output* $Y_N$, *it holds that*

$$\mathbb{E}[\mathrm{KL}(\mu, \hat{p})] \lesssim \epsilon \beta^{d+3}(b+d)/a^2 \,,$$

*where* $\lesssim$ *hides a universal constant as well as log factors with respect to* $d, \epsilon^{-1}, a, b, \beta$. *In particular, the error above is achieved in* $\sum_{k=1}^N n_k = d\epsilon^{-2(d+2)}$ *queries to* $V$.

The proof is deferred to Appendix F and is mainly an application of the results collected above.

# 6 Conclusion

In this article, we successfully applied the reduction from sampling to intermediate score estimation, initiated over the past three years by Chen et al. [2023b,a], Conforti et al. [2025], Benton et al. [2024], to the problem of low dimensional multi-modal sampling. Using the self-normalized estimator of the scores, our results provide *polynomial* query complexity guarantees in fixed dimension, apply to *general* Gaussian mixtures, and do *not* require prior knowledge of the target distribution's constants. Interesting future directions include extending theoretical guarantees to more general multi-modal distributions with heavy tails for instance.

We note that our sampling algorithm is based on time-reversed diffusions, which have recently gained traction for sampling from unnormalized densities. Our method stands out by offering rigorous, non-asymptotic theoretical guarantees on query complexity, especially in the presence of score estimation error that we precisely quantify. Such theoretical guarantees are scarce and we believe our results are therefore a meaningful and timely contribution to the field.

# 7 Acknowledgements

This work was supported by the Agence Nationale de la Recherche (ANR) through the JCJC WOS project and the PEPR PDE-AI project (ANR-23-PEIA-0004). We thank Pierre Monmarché for his help in proving Lemma 13.

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

# Appendix  Table of contents

# A   Expressions the intermediate scores

We next provide derivations of the Monte Carlo estimators of the intermediate scores discussed in section 4.2.

The probability law along the reverse diffusion path has density

$$p_t^V(x) = \frac{1}{\sqrt{1-\lambda_t}}\pi\left(\frac{x}{\sqrt{1-\lambda_t}}\right) * \frac{1}{\sqrt{\lambda_t}}\mu\left(\frac{x}{\sqrt{\lambda_t}}\right) \tag{6}$$

where $\pi$ is a standard Gaussian and $\mu(\cdot) \propto \exp(-V(\cdot))$ is the target distribution

**Case 1: write the convolution as an integral against the proposal distribution**   We have

$$p_t^V(x) = \frac{1}{Z_{\lambda_t}}\int_{y\in\mathbb{R}^d}\mu\left(\frac{x-y}{\sqrt{\lambda_t}}\right)\pi\left(\frac{y}{\sqrt{1-\lambda_t}}\right)dy = \frac{1}{Z_{\lambda_t}}\int f(x,y)dy \tag{7}$$

where we denoted the integrand by $f(x,y)$. We can compute the score

$$\nabla\log p_t^V(x) = \frac{\nabla p_t^V(x)}{p_t^V(x)} = \frac{\int\nabla f(x,y)dy}{\int f(x,y)dy} = \frac{1}{\sqrt{\lambda_t}}\int\nabla\log\mu\left(\frac{x-y}{\sqrt{\lambda_t}}\right)\frac{f(x,y)}{\int f(x,y)dy}dy \ . \tag{8}$$

From this, we can either define a Monte Carlo estimator as

$$\nabla\log p_t^V(x) = \frac{1}{\sqrt{\lambda_t}}\mathbb{E}_{y|x}\left[\nabla\log\mu\left(\frac{x-y}{\sqrt{\lambda_t}}\right)\right], \quad y|x \sim\propto f(x,y) \tag{9}$$

where $f(x,y)$ is a smoothened version of the target distribution distribution. Replacing $\pi$ with a standard Gaussian. Or else, by unpacking $f(x,y)$ and using the proposal as the sampling distribution as the integration variable $y$ appears in it,

$$\nabla\log p_t^V(x) = \frac{1}{\sqrt{\lambda_t}}\frac{\mathbb{E}_y\left[\nabla\log\mu\left(\frac{x-y}{\sqrt{\lambda_t}}\right)\mu\left(\frac{x-y}{\sqrt{\lambda_t}}\right)\right]}{\mathbb{E}_y[\mu\left(\frac{x-y}{\sqrt{\lambda_t}}\right)]}, \quad y \sim \pi\left(\frac{y}{\sqrt{1-\lambda_t}}\right) \tag{10}$$

**Case 2: write the convolution as an integral against the target distribution**   We now have

$$p_t^V(x) = \frac{1}{Z_{\lambda_t}}\int_{y\in\mathbb{R}^d}\mu\left(\frac{y}{\sqrt{\lambda_t}}\right)\pi\left(\frac{x-y}{\sqrt{1-\lambda_t}}\right)dy = \frac{1}{Z_{\lambda_t}}\int f(x,y)dy \tag{11}$$

where we now denote the integrand by $f(x,y)$. We can compute the score

$$\nabla\log p_t^V(x) = \frac{\nabla p_t^V(x)}{p_t^V(x)} = \frac{\int\nabla f(x,y)dy}{\int f(x,y)dy} \tag{12}$$

$$= \frac{1}{\sqrt{1-\lambda_t}}\int\nabla\log\pi\left(\frac{x-y}{\sqrt{1-\lambda_t}}\right)\frac{f(x,y)}{\int f(x,y)dy}dy \ . \tag{13}$$

From this, we can either define a Monte Carlo estimator as

$$\nabla\log p_t^V(x) = \frac{1}{\sqrt{1-\lambda_t}}\mathbb{E}_{y|x}\left[\nabla\log\pi\left(\frac{x-y}{\sqrt{1-\lambda_t}}\right)\right], \quad y|x \sim\propto f(x,y) \tag{14}$$

where $f(x,y)$ is a smoothened version of the target distribution distribution. Or else, by upacking $f(x,y)$ and using the target as the sampling distribution as the integration variable $y$ appears in it,

$$\nabla\log p_t^V(x) = \frac{1}{\sqrt{1-\lambda_t}}\frac{\mathbb{E}_y\left[\nabla\log\pi\left(\frac{x-y}{\sqrt{1-\lambda_t}}\right)\pi\left(\frac{x-y}{\sqrt{1-\lambda_t}}\right)\right]}{\mathbb{E}_y[\pi\left(\frac{x-y}{\sqrt{1-\lambda_t}}\right)]}, \quad y \sim \mu\left(\frac{y}{\sqrt{\lambda_t}}\right) \tag{15}$$

which is not very useful given that sampling the target is hard in the first place. However, we can use the proposal distribution as the sampling distribution as the integration variable $y$ appears in it as well. To see this, recall that

$$\pi\left(\frac{x-y}{\sqrt{1-\lambda_t}}\right) = \mathcal{N}\left(\frac{x-y}{\sqrt{1-\lambda_t}};0,I_d\right) = \sqrt{1-\lambda_t}\mathcal{N}\left(x-y;0,(1-\lambda_t)I_d\right) \tag{16}$$

$$= \sqrt{1-\lambda_t}\mathcal{N}\left(y;x,(1-\lambda_t)I_d\right) \tag{17}$$

This leads to

$$\nabla \log p_t^V(x) = \frac{1}{1-\lambda_t} \frac{\mathbb{E}_y\left[(y-x)\mu\left(\frac{y}{\sqrt{\lambda_t}}\right)\right]}{\mathbb{E}_y\left[\mu\left(\frac{y}{\sqrt{\lambda_t}}\right)\right]}, \quad y \sim \mathcal{N}(y; x, (1-\lambda_t)I_d) \tag{18}$$

$$= \frac{1}{\sqrt{1-\lambda_t}} \frac{\mathbb{E}_z\left[z\mu\left(\frac{x+\sqrt{1-\lambda_t}z}{\sqrt{\lambda_t}}\right)\right]}{\mathbb{E}_z\left[\mu\left(\frac{x+\sqrt{1-\lambda_t}z}{\sqrt{\lambda_t}}\right)\right]}, \quad z \sim \mathcal{N}(z; 0, I_d). \tag{19}$$

Using a change of variables, we obtain

$$\nabla \log\left(p_t^V\right)(x) = \frac{-1}{1-\lambda_t} \frac{\mathbb{E}\left[Y_t \exp\left(-V\left(\frac{x-Y_t}{\sqrt{\lambda_t}}\right)\right)\right]}{\mathbb{E}\left[\exp\left(-V\left(\frac{x-Y_t}{\sqrt{\lambda_t}}\right)\right)\right]} \tag{20}$$

with $Y_t \sim \mathcal{N}(0, (1-\lambda_t)I_d)$.

# B Additional discussions

## B.1 Föllmer Flows

### B.1.1 Comparison with our work

Instead of implementing a reverse diffusion process, Föllmer Flows Huang et al. [2025], Jiao et al. [2021], Ruzayqat et al. [2023] seek to implement the following Schrödinger bridge:

$$\mathrm{d}X_t = b(X_t, t)\mathrm{d}t + \mathrm{d}W_t, \quad X_0 = 0,$$

with $b(x, t) = \nabla \log \mathbb{E}[f(x + W_{1-t})]$ and where $f$ is given by

$$f := \frac{\mathrm{d}\mu}{\mathrm{d}\mathcal{N}(0, I_d)},$$

with $\mu$ the target density; for such a process, we have that $X_1 \sim \mu$. We refer the reader to the notations of Proposition 14 to observe that the shift $b(x, t)$ is given by

$$b(x, t) = \nabla \log Q_{-\log(t)/2}(x/\sqrt{t}),$$
$$= \frac{1}{\sqrt{t}}(\nabla \log p^V_{-\log(t)/2}(x/\sqrt{t}) - \nabla \log \pi(x)),$$

with $\pi$ is the density of the standard Gaussian. Hence, up to the Gaussian term and a time-rescaling, the shift is exactly given by the intermediate scores of the reverse diffusion. However, because of the Gaussian correction, an extra exponential term appears in the associated self-normalized estimator of the shift: in Huang et al. [2025], Jiao et al. [2021], Ruzayqat et al. [2023], the authors implement

$$\hat{b}(x, t) = \frac{\sum_{i=1}^n (x + \sqrt{1-t}z_i - \nabla V(x + \sqrt{1-t}z_i))e^{-V(x+\sqrt{1-t}z_i)+\|x+\sqrt{1-t}z_i\|^2/2}}{\sum_{i=1}^n e^{-V(x+\sqrt{1-t}z_i)+\|x+\sqrt{1-t}z_i\|^2/2}},$$

with $z_i$ $n$-i.i.d. samples from the standard Gaussian distribution. We suspect that because of the extra exponential terms $e^{\|x+\sqrt{1-t}z_i\|^2/2}$, as discussed below, their estimator provide appealing guarantees only under very stringent assumptions.

### B.1.2 Theoretical guarantees

The theoretical complexity bounds derived in the works of Huang et al. [2025], Jiao et al. [2021], Ruzayqat et al. [2023] quantitatively rely on the assumptions that $f = \frac{\mathrm{d}\mu}{\mathrm{d}\mathcal{N}(0, I_d)}$ is Lipschitz, has Lipschitz gradient and is bounded from below. In particular, assume for instance that $\mu$ is a standard Gaussian centered at some point $c \in \mathbb{R}^d$. In this case, the ratio $f$ reads

$$f(x) = e^{2x^\top c + \|c\|^2},$$

which verifies none of the assumptions above if $c \neq 0$. More broadly, even if $f$ verifies these assumptions, the resulting quantities degrade exponentially with the mismatch between $x \mapsto -\log(\mu)(x)$ and $x \mapsto \|x\|^2/2$. As noted in Vargas et al. [2022], this limitation is not only a theoretical artifact; in practice, these methods are very unstable and fail to convergence on simple examples.

## B.2 Details on the non-smooth example

Consider the mixture $\mu = 0.5\mathcal{N}(0, \Sigma_1) + 0.5\mathcal{N}(0, \Sigma_2)$ with $\Sigma_1 = diag(1, 0.5)$ and $\Sigma_1 = diag(0.5, 1)$. For $(x, y) \in \mathbb{R}^2$, it holds that

$$-\nabla \log(\mu)(x, y) = \frac{(x, 2y)e^{-x^2/2-y^2} + (2x, y)e^{-x^2-y^2/2}}{e^{-x^2/2-y^2} + e^{-x^2-y^2/2}}.$$

Hence, when $x = y$, we have $-\nabla \log(\mu)(x, y) = 3/2(x, x)$. Now for $y = x + \eta$, the score reads

$$-\nabla \log(\mu)(x, x + \eta) = \frac{(x, 2x + 2\eta)e^{-x^2/2-x^2-2\eta x-\eta^2} + (2x, x + \eta)e^{-x^2-x^2/2-\eta x-\eta^2/2}}{e^{-x^2/2-x^2-2\eta x-\eta^2} + e^{-x^2-x^2/2-\eta x-\eta^2/2}}$$
$$= \frac{(x, 2x + 2\eta)e^{-\eta x-\eta^2/2} + (2x, x + \eta)}{e^{-\eta x-\eta^2/2} + 1}$$
$$\sim_{x \to +\infty} (2x, x).$$

In particular, $-\nabla \log(\mu)$ is not Hölder.

### B.3 Proof of Proposition 2

Recall we denote $\lambda_{I_d n} := I_d n_i \lambda_{I_d n}(\Sigma_i), \quad \lambda_{\max} := \max_i \lambda_{\max}(\Sigma_i)$. We write:

$$\mu(x) = \sum_{i=1}^{p} \tilde{w}_i \exp\left(-\frac{1}{2}(x - \mu_i)^\top \Sigma_i^{-1}(x - \mu_i)\right) := \sum_{i=1}^{p} \tilde{w}_i \phi_i(x)$$

with $\tilde{w}_i = w_i (2\pi)^{-d/2} \det(\Sigma_i)^{-1/2}$.

**Bound on the Hessian.** The Hessian of $\mu$ writes:

$$\nabla^2 \log \mu(x) = \frac{1}{\mu(x)} \nabla^2 \mu(x) - \frac{1}{\mu(x)^2} \nabla \mu(x) \nabla \mu(x)^\top$$

Since

$$\nabla \phi_i(x) = -\Sigma_i^{-1}(x - \mu_i)\, \phi_i(x)$$
$$\nabla^2 \phi_i(x) = \left[\Sigma_i^{-1}(x - \mu_i)(x - \mu_i)^\top \Sigma_i^{-1} - \Sigma_i^{-1}\right] \phi_i(x)$$

We have

$$\nabla \mu(x) = \sum_i \tilde{w}_i \nabla \phi_i(x) = -\sum_i \tilde{w}_i \phi_i(x) \Sigma_i^{-1}(x - \mu_i)$$

$$\nabla^2 \mu(x) = \sum_{i=1}^{p} \tilde{w}_i \nabla^2 \phi_i(x) = \sum_{i=1}^{p} \tilde{w}_i \phi_i(x) \left[\Sigma_i^{-1}(x - \mu_i)(x - \mu_i)^\top \Sigma_i^{-1} - \Sigma_i^{-1}\right]$$

Denoting $\gamma_i(x) := \frac{\tilde{w}_i \phi_i(x)}{\mu(x)}$ and $s_i(x) = -\Sigma_i^{-1}(x - \mu_i)$ we get

$$\nabla^2 \log \mu(x) = \sum_i \gamma_i(x) \left[s_i(x) s_i(x)^\top - \Sigma_i^{-1}\right] - \left(\sum_i \gamma_i(x) s_i(x)\right)\left(\sum_j \gamma_j(x) s_j(x)\right)^\top$$

$$= \mathrm{Cov}_{\gamma(x)}[s_i(x)] - \sum_i \gamma_i(x) \Sigma_i^{-1}$$

where the first term is a covariance matrix of the vectors $s_i(x)$ under the weights $\gamma_i(x)$, hence which is a positive semi-definite matrix. Therefore:

$$-\nabla^2 \log \mu(x) \preceq \sum_i \gamma_i(x) \Sigma_i^{-1} \preceq \sum_i \gamma_i(x) \cdot \frac{I_d}{\lambda_{I_d n}} = \frac{I_d}{\lambda_{I_d n}}.$$

**Bound on the drift.** For the (negative) score of the mixture we have:

$$-\nabla \log \mu(x) = \sum_i \gamma_i(x) \Sigma_i^{-1}(x - \mu_i)$$

Hence

$$\langle -\nabla \log \mu(x), x \rangle = \sum_{i=1}^{p} \gamma_i(x) \langle \Sigma_i^{-1}(x - \mu_i), x \rangle$$

$$= \sum_{i=1}^{p} \gamma_i(x) \left(x^\top \Sigma_i^{-1} x - \mu_i^\top \Sigma_i^{-1} x\right) \geq \sum_{i=1}^{p} \gamma_i(x) \left(\frac{\|x\|^2}{\lambda_{\max}} - \mu_i^\top \Sigma_i^{-1} x\right),$$

where we have used $\Sigma_i^{-1} \succeq \frac{1}{\lambda_{\max}} I_d$ for the first term. Now for the second term, using Cauchy-Schwartz and Young's inequality:

$$\mu_i^\top \Sigma_i^{-1} x \leq \|\mu_i\| \cdot \|\Sigma_i^{-1} x\| \leq \frac{\|\mu_i\| \|x\|}{\lambda_{I_d n}} \leq \frac{1}{2\lambda_{I_d n}}(\|x\|^2 + \|\mu_i\|^2).$$

Hence, using that $\sum_i \gamma_i(x) = 1$ and that $\sum_i \gamma_i(x)\|\mu_i\|^2 \le \max_i \|\mu_i\|^2$ we have:

$$\langle -\nabla \log \mu(x), x \rangle \ge \sum_{i=1}^{p} \gamma_i(x) \left( \frac{\|x\|^2}{\lambda_{\max}} - \frac{1}{2\lambda_{I_d n}}(\|x\|^2 + \|\mu_i\|^2) \right)$$

$$= \left( \frac{1}{\lambda_{\max}} - \frac{1}{2\lambda_{I_d n}} \right) \|x\|^2 - \frac{1}{2\lambda_{I_d n}} \sum_{i=1}^{p} \gamma_i(x)\|\mu_i\|^2.$$

$$\ge \left( \frac{1}{\lambda_{\max}} - \frac{1}{2\lambda_{I_d n}} \right) \|x\|^2 - \frac{1}{2\lambda_{I_d n}} \max_i \|\mu_i\|^2$$

$$\ge \frac{\|x\|^2}{2\lambda_{\max}} - \lambda_{\max} \cdot \max_i \left( \frac{\|\mu_i\|}{\lambda_{I_d n}} \right)^2.$$

## B.4 Discussion on Lytras and Mertikopoulos [2025]

To justify our claims at the end of Section 3.2, we first need some preliminary background on Poincaré constants. Let $q \in \mathcal{P}_{ac}(\mathbb{R}^d)$. We say that $q$ satisfies the *Poincaré inequality* with constant $C_P \ge 0$ if for all $f \in W_0^{1,2}(q)$ (functions with zero-mean with respect to $q$ and whose gradient is squared integrable with respect to $q$):

$$\int f^2(x) dq(x) \le C_P \|\nabla f\|_{L^2(q)}^2, \tag{21}$$

and let $C_P(q)$ be the best constant in Eq. 21, or $+\infty$ if it does not exist. We show that in the case where $\mu = 1/2\mathcal{N}(c, \lambda_{max}) + 1/2\mathcal{N}(-c, \lambda_{min})$ the Poincaré constant is at least $2e^{\frac{\|c\|^2}{2\lambda_{max}}}/\|c\|^2$ therefore the resulting sampling complexity in Lytras and Mertikopoulos [2025, Theorem 3] is at least $O(\text{poly}(\|c\|^2 e^{\frac{\|c\|^2}{2\lambda_{max}}}, \epsilon^{-1}))$.

**Proposition 10** *Let* $\mu = 1/2\mathcal{N}(c, \lambda_{max} I_d) + 1/2\mathcal{N}(-c, \lambda_{min} I_d)$ *with* $\lambda_{max} \ge \lambda_{min} > 0$ *and* $\|c\| > 0$. *It holds that*

$$C_P(\mu) \ge \frac{\|c\|^2 e^{\frac{\|c\|^2}{2\lambda_{max}}}}{2}.$$

*Proof.* Denoting $u = c/\|c\|$ and defining $g : \mathbb{R} \to \mathbb{R}$ as

$$\begin{cases} g(t) = -1 \text{ if } t < -\|c\|/2, \\ g(t) = 2t/\|c\| \text{ if } -\|c\|/2 \le t \le \|c\|/2, \\ g(t) = 1 \text{ if } t > \|c\|/2, \end{cases}$$

consider the test function $f(x) = g(x^\top u)$. Denoting $\mu_1 = \mathcal{N}(c, \lambda_{max} I_d)$, $\mu_2 = \mathcal{N}(-c, \lambda_{min} I_d)$, we start to upper-bound the Dirichlet energy $I = \int \|\nabla f\|^2 d\mu$:

$$I = \frac{2}{\|c\|^2} \left( \mathbb{P}_{X \sim \mu_1} \left[ -\frac{\|c\|}{2} \le X^\top u \le \frac{\|c\|}{2} \right] + \mathbb{P}_{X \sim \mu_2} \left[ -\frac{\|c\|}{2} \le X^\top u \le \frac{\|c\|}{2} \right] \right).$$

Now remark that for $X \sim \mu_i$, the random variable $X^\top u$ is also a one dimensional gaussian: if $X \sim \mu_1$, it holds that $X^\top u \sim \mathcal{N}(\|c\|, \lambda_{max})$ and if $X \sim \mu_2$, it holds that $X^\top u \sim \mathcal{N}(-\|c\|, \lambda_{min})$. In particular, the Dirichlet energy re-writes as

$$I = \frac{2}{\|c\|^2} \left( \Phi(\frac{-\|c\|}{2\sqrt{\lambda_{max}}}) - \Phi(\frac{-3\|c\|}{2\sqrt{\lambda_{max}}}) + \Phi(\frac{-\|c\|}{2\sqrt{\lambda_{min}}}) - \Phi(\frac{-3\|c\|}{2\sqrt{\lambda_{min}}}) \right),$$

where $\Phi$ is the cumulative density function of the standard Gaussian. In particular, $I$ is upper-bounded as

$$I \le \frac{4\Phi(\frac{-\|c\|}{2\sqrt{\lambda_{max}}})}{\|c\|^2} = \frac{2}{\|c\|^2}(1 - \text{erf}(\frac{\|c\|}{\sqrt{2\lambda_{max}}})) \le \frac{2e^{-\frac{\|c\|^2}{2\lambda_{max}}}}{\|c\|^2}.$$

Now there remains to lower-bound the variance term. The variance reads

$$\text{Var}_\mu(f) = \int f^2 d\mu = 1/2(\mathbb{E}_{\mu_1}[f^2(X)] + \mathbb{E}_{\mu_2}[f^2(X)]).$$

The first term can be lower-bounded as

$$\mathbb{E}_{\mu_1}[f^2(X)] \geq \mathbb{P}_{X \sim \mu_1}[X^\top \mu \geq \frac{\|c\|}{2}],$$

$$= \Phi(\frac{\|c\|}{2\sqrt{\lambda_{\max}}}),$$

$$\geq 1/2.$$

Conversely, it holds that $\mathbb{E}_{\mu_2}[f^2(X)] \geq 1/2$ and *a fortiori* $\operatorname{Var}_\mu(f) \geq 1$. Hence we recover that

$$\frac{\operatorname{Var}_\mu(f)}{I} \geq \frac{\|c\|^2 e^{\frac{\|c\|^2}{2\lambda_{max}}}}{2},$$

which concludes the proof. □

## B.5 Discussion on Agapiou et al. [2017]

In Agapiou et al. [2017], for $\nu$ dominated by $\pi$, denoting $g$ the unnormalised density ratio

$$\frac{\mathrm{d}\nu}{\mathrm{d}\pi}(u) := \frac{g(u)}{\int g(u)\mathrm{d}\pi(u)},$$

the authors derived error bounds for *self-normalised* estimators: for any test function $\phi : \mathbb{R}^d \to \mathbb{R}$, the expectation of $\phi$ under $\nu$, denoted $\nu(\phi)$, is estimated via

$$\nu^n(\phi) = \frac{\sum_{i=1}^n \phi(x_i)g(x_i)}{\sum_{i=1}^n g(x_i)},$$

with $(x_i)$ $n$-iid samples drawn from $\pi$. In Theorem 2.3, they provide the following quadratic error bound:

$$\mathbb{E}[(\nu(\phi) - \nu^n(\phi))^2] \lesssim \frac{3\pi(|\phi g|^{2d})^{1/d} m_{2e}(g)^{1/e}}{n\pi(g)^4}, \tag{22}$$

with $m_t(h) = \pi(|h(\cdot) - \pi(h)|^t)$ and $d,e \in (1,+\infty[$ such that $1/d + 1/e = 1$. Now recall that, denoting $Y_t \sim \mathcal{N}(0, (1 - e^{-2t})I_d)$ the score along the forward is given by

$$\nabla \log(p_t^V)(z) = \frac{-1}{1 - e^{-2t}} \frac{\mathbb{E}[Y_t e^{-V(e^t(z-Y_t))}]}{\mathbb{E}[e^{-V(e^t(z-Y_t))}]},$$

$$= \frac{-1}{1 - e^{-2t}} \frac{\int y e^{-V(e^t(z-y))} e^{-\frac{\|y\|^2}{2(1-e^{-2t})}}\mathrm{d}y}{\int e^{-V(e^t(z-y))} e^{-\frac{\|y\|^2}{2(1-e^{-2t})}}\mathrm{d}y},$$

$$= \frac{-1}{1 - e^{-2t}} \mathbb{E}_{Z \sim \nu}[Z],$$

where $\nu(x) \propto e^{-V(e^t(z-x))} e^{-\frac{\|x\|^2}{2(1-e^{-2t})}}$. In particular, denoting $\pi(x) \propto e^{-\frac{\|x\|^2}{2(1-e^{-2t})}}$, the quadratic error of the self-normalized estimator reads

$$\|\hat{s}_{t,n}(z) - \nabla \log p_t^V(z)\|^2 = \frac{1}{(1 - e^{-2t})^2} \left\| \frac{\sum_{i=1}^n y_i g(y_i)}{\sum_{i=1}^n g(y_i)} - \mathbb{E}_{Z \sim \nu}[Z] \right\|^2,$$

$$= \frac{1}{(1 - e^{-2t})^2} \sum_{k=1}^d \left( \frac{\sum_{i=1}^n \phi_k(y_i)g(y_i)}{\sum_{i=1}^n g(y_i)} - \mathbb{E}_{Z \sim \nu}[\phi_k(Z)] \right)^2.$$

with $g = \mathrm{d}\nu/\mathrm{d}\pi$, $(y_i)$ $n$-iid samples drawn from $\pi$ and $\phi_k : \mathbb{R}^d \to \mathbb{R}$ the projector on the $k$-th coordinate. Hence, the bound Eq. 22 of Agapiou et al. [2017] provides the following upper-bound

$$\|\hat{s}_{t,n}(z) - \nabla \log p_t^V(z)\|^2 \leq \sum_{k=1}^d \frac{3\pi(|\phi_k g|^{2d})^{1/d} m_{2e}(g)^{1/e}}{n\pi(g)^4}.$$

We show in the next paragraph that, when integrated against $p_t^V$, this upper-bound is vacuous. By Jensen's inequality, it holds that

$$\begin{cases} m_{2e}(g)^{1/(2e)} \geq m_2(g)^{1/2}\,, \\ \pi(|\phi_k g|^{2d})^{1/(2d)} \geq \pi(|\phi_k g|^2)^{1/2}\,. \end{cases}$$

Hence, the previous right-hand side of is lower-bounded as

$$\sum_{k=1}^d \frac{3\pi(|\phi_k g|^{2d})^{1/d} m_{2e}(g)^{1/e}}{n\pi(g)^4} \geq \sum_{k=1}^d \frac{3\pi(|\phi_k g|^2) m_2(g)}{n\pi(g)^4} = \frac{3m_2(g)}{n\pi(g)^4} \sum_{k=1}^d \pi(|\phi_k g|^2)\,.$$

We now compute explicitly each of the quantities in the right-hand-side above using Proposition 11. First we have

$$\pi(g) = (2\pi(1 - e^{-2t}))^{-d/2} \int e^{-V(e^t(z-x))} e^{-\frac{\|x\|^2}{2(1-e^{-2t})}}\,\mathrm{d}x\,,$$

$$= e^{-td} Z_V p_t^V(z)\,.$$

Then, the variance term reads

$$m_2(g) = (2\pi(1 - e^{-2t}))^{-d/2} \int (e^{-V(e^t(z-x))} - e^{-td} Z_V p_t^V(z))^2 e^{-\frac{\|x\|^2}{2(1-e^{-2t})}}\,\mathrm{d}x\,,$$

$$= e^{-td} Z_{2V} p_t^{2V}(z) - 2e^{-2td} Z_{2V} Z_V p_t^V(z) p_t^{2V}(z) + (e^{-td} Z_V)^2 p_t^V(z)^2\,.$$

Finally, the second order moment term reads

$$\sum_{k=1}^d \pi(|\phi_k g|^2) = (2\pi(1 - e^{-2t}))^{-d/2} \int \|x\|^2 e^{-2V(e^t(z-x))} e^{-\frac{\|x\|^2}{2(1-e^{-2t})}}\,\mathrm{d}x\,,$$

$$= e^{-td} p_t^{2V}(z) \Theta(z)\,,$$

where we denoted

$$\Theta(z) := Z_{2V}(1 - e^{-2t})^2 (\Delta \log(p_t^{2V})(z) + \|\nabla \log(p_t^{2V})(z)\|^2 + d/(1 - e^{-2t}))\,.$$

Hence, once integrated against $p_t^V$, our lower bound reads

$$\int \frac{3m_2(g)}{n\pi(g)^4} \sum_{k=1}^d \pi(|\phi_k g|^2) p_t^V(z)\mathrm{d}z = e^{2td} \frac{Z_{2V}^2}{Z_V^4} \int \Theta(z) \frac{p_t^{2V}(z)^2}{p_t^V(z)^3}\mathrm{d}z - 2e^{td} \frac{Z_{2V}^2}{Z_V^3} \int \Theta(z) \frac{p_t^{2V}(z)^2}{p_t^V(z)^2}\mathrm{d}z$$

$$+ e^{td} \frac{Z_{2V}}{Z_V^2} \int \Theta(z) \frac{p_t^{2V}(z)}{p_t^V(z)}\mathrm{d}z\,.$$

Combining Proposition 6 and Lemma 7, we have that the third term is always finite. For the specific case $V(x) = \frac{\|x\|^2}{2}$, we show that the second term also remains finite while the first diverges thus making the overall lower bound diverge. First, we recall that for this choice of potential, we have

$$\begin{cases} p_t^V(z) \propto e^{-\frac{\|z\|^2}{2}}\,, \\ p_t^{2V}(z) \propto e^{-\frac{\|z\|^2}{2(1-e^{-2t}/2)}}\,, \end{cases}$$

from what we obtain

$$\begin{cases} \Theta(z) = Z_{2V}\left(\|z\|^2 + \frac{d(1-e^{-2t})[2(1-e^{-2t})+e^{-4t}]}{2-e^{-2t}}\right)\,, \\ \frac{p_t^{2V}(z)^2}{p_t^V(z)^2} \propto e^{-\frac{\|e^{-t}z\|^2}{2(1-e^{-2t}/2)}}\,, \\ \frac{p_t^{2V}(z)^2}{p_t^V(z)^3} \propto e^{\frac{\|z\|^2(1-e^{-2t})}{2(1-e^{-2t}/2)}}\,. \end{cases}$$

In particular, we obtain that the second term is bounded and that the first term diverges to infinity making the overall bound of Agapiou et al. [2017] vacuous.

## C  Proof of Proposition 4

Before starting the proof, we recall the following identities.

**Proposition 11** (Tweedie's formulas) *Denoting $p_t^V$ the density of the forward process $X_t$ initialized at $\mu \propto e^{-V}$, and $Y_t \sim \mathcal{N}(0, (1 - e^{-2t})I_d)$, it holds for all $z \in \mathbb{R}^d$ that*

$$p_t^V(z) = \frac{e^{td}}{Z_V}\mathbb{E}[e^{-V((z-Y_t)e^t)}], \tag{23}$$

*that*

$$\nabla \log(p_t^V)(z) = \frac{\mathbb{E}[-Y_t e^{-V((z-Y_t)e^t)}]}{(1-e^{-2t})\mathbb{E}[e^{-V((z-Y_t)e^t)}]}, \tag{24}$$

*and that*

$$\nabla^2 \log(p_t^V)(z) = \frac{\mathbb{E}[Y_t Y_t^\top e^{-V((z-Y_t)e^t)}]}{(1-e^{-2t})^2\mathbb{E}[e^{-V((z-Y_t)e^t)}]} - \frac{I_d}{(1-e^{-2t})} \tag{25}$$
$$- (\nabla \log(p_t^V)(z)))(\nabla \log(p_t^V)(z))^\top.$$

*Proof.* Recall that $p_t^V$ is the law of the variable

$$X_t = e^{-t}X_0 + B_{1-e^{-2t}},$$

with $X_0 \sim \mu$ and $B_s$ the standard Brownian motion evaluated at time $s$. Hence, using Bayes formula, we have

$$p_t^V(z) = \int p_t(z|x)dp_0(x) = \frac{1}{Z^V(1-e^{-2t})^{d/2}(2\pi)^{d/2}}\int_{\mathbb{R}^d} e^{-\frac{\|z-xe^{-t}\|^2}{2(1-e^{-2t})}}e^{-V(x)}dx.$$

After taking the logarithm and differentiating with respect to $z$, we obtain

$$\nabla \log(p_t^V)(z) = \frac{\int_{\mathbb{R}^d} -(z-xe^{-t})e^{-\frac{\|z-xe^{-t}\|^2}{2(1-e^{-2t})}}e^{-V(x)}dx}{(1-e^{-2t})\int_{\mathbb{R}^d} e^{-\frac{\|z-xe^{-t}\|^2}{2(1-e^{-2t})}}e^{-V(x)}dx}.$$

To obtain the Hessian, we differentiate the formula above. The Jacobian of the numerator is given by

$$-I_d \int_{\mathbb{R}^d} e^{-\frac{\|z-xe^{-t}\|^2}{2(1-e^{-2t})}}e^{-V(x)}dx + \frac{1}{1-e^{-2t}}\int_{\mathbb{R}^d}(z-xe^{-t})(z-xe^{-t})^\top e^{-\frac{\|z-xe^{-t}\|^2}{2(1-e^{-2t})}}e^{-V(x)}dx,$$

from which we can deduce

$$\nabla^2 \log(p_t^V)(z) = -\frac{I_d}{(1-e^{-2t})} + \frac{\int_{\mathbb{R}^d}(z-xe^{-t})(z-xe^{-t})^\top e^{-\frac{\|z-xe^{-t}\|^2}{2(1-e^{-2t})}}e^{-V(x)}dx}{(1-e^{-2t})^2\int_{\mathbb{R}^d} e^{-\frac{\|z-xe^{-t}\|^2}{2(1-e^{-2t})}}e^{-V(x)}dx}$$
$$- \frac{(\int_{\mathbb{R}^d}(z-xe^{-t})e^{-\frac{\|z-xe^{-t}\|^2}{2(1-e^{-2t})}}e^{-V(x)}dx)(\int_{\mathbb{R}^d}(z-xe^{-t})e^{-\frac{\|z-xe^{-t}\|^2}{2(1-e^{-2t})}}e^{-V(x)}dx)^\top}{(1-e^{-2t})^2(\int_{\mathbb{R}^d} e^{-\frac{\|z-xe^{-t}\|^2}{2(1-e^{-2t})}}e^{-V(x)}dx)^2}$$
$$= -\frac{I_d}{(1-e^{-2t})} + \frac{\int_{\mathbb{R}^d}(z-xe^{-t})(z-xe^{-t})^\top e^{-\frac{\|z-xe^{-t}\|^2}{2(1-e^{-2t})}}e^{-V(x)}dx}{(1-e^{-2t})^2\int_{\mathbb{R}^d} e^{-\frac{\|z-xe^{-t}\|^2}{2(1-e^{-2t})}}e^{-V(x)}dx}$$
$$- (\nabla \log(p_t^V)(z))(\nabla \log(p_t^V)(z))^\top.$$

In order to rewrite the quantities above as expectations, we make the change of variable $y = z - xe^{-t}$ so that $x = (z-y)e^t$ and we obtain for the density $p_t^V$:

$$p_t^V(z) = \frac{e^{td}}{Z_V}\mathbb{E}[e^{-V((z-Y_t)e^t)}],$$

where $Y_t \sim \mathcal{N}(0, I_d(1 - e^{-2t}))$. Conversely, the score rewrites as

$$\nabla \log(p_t^V)(z) = \frac{\mathbb{E}[-Y_t e^{-V((z-Y_t)e^t)}]}{(1 - e^{-2t})\mathbb{E}[e^{-V((z-Y_t)e^t)}]},$$

and the Hessian rewrites as

$$\nabla^2 \log(p_t^V)(z) = \frac{\mathbb{E}[Y_t Y_t^\top e^{-V((z-Y_t)e^t)}]}{(1 - e^{-2t})^2 \mathbb{E}[e^{-V((z-Y_t)e^t)}]} - \frac{I_d}{(1 - e^{-2t})} - (\nabla \log(p_t^V)(z))(\nabla \log(p_t^V)(z))^\top.$$

$\square$

For the rest of the proof we shall drop the dependence in $z$ and write $\hat{s}_{t,n}(z) = \frac{\hat{N}}{\hat{D}}$ with the empirical numerator $\hat{N} = -\frac{\sum_{i=1}^n y_i e^{-V(e^t(z-y_i))}}{1-e^{-2t}}$ and denominator $\hat{D} = \sum_{i=1}^n e^{-V(e^t(z-y_i))}$ where we recall $y_i \sim \mathcal{N}(0, (1 - e^{-2t})I_d)$. In the following proposition, we explicitly compute the variances of $\hat{N}$ and $\hat{D}$ using the formulas above. In what follows, we shall denote $N = \mathbb{E}[N]$ and $D = \mathbb{E}[D]$

**Proposition 12** (Variance of estimators) *Let $y_1, \ldots, y_n$ i.i.d. distributed as $\mathcal{N}(0, (1 - e^{-2t})I_d)$. Denote by $\pi$ a standard normal density, and by $\hat{N}(z)$ and $\hat{D}(z)$ the numerator and denominator of the estimator defined in Eq. 5. We have:*

$$\mathbb{E}[\|\hat{N} - N\|^2] \leq \frac{p_t^{2V} Z_{2V} e^{-td}}{n}\left(\Delta \log(p_t^{2V})(z) - \frac{\Delta \log(\pi)(z)}{1 - e^{-2t}} + \|\nabla \log(p_t^{2V})\|^2(z)\right),$$

$$\mathbb{E}[\|\hat{D} - D\|^2] \leq \frac{p_t^{2V}(z) Z_{2V} e^{-td}}{n}.$$

*Proof.* For the numerator, we have

$$\hat{N} - N = \frac{-1}{n}\sum_{i=1}^n \frac{y_i e^{-V((z-y_i)e^t)}}{1 - e^{-2t}} + N,$$

hence, since the $y_i$, $i = 1, \ldots, n$ are i.i.d. distributed as $Y_t \sim \mathcal{N}(0, (1 - e^{-2t})I_d)$,

$$\mathbb{E}[\|\hat{N} - N\|^2] = \frac{1}{n}\mathbb{E}\left[\left\|\frac{Y_t e^{-V((z-Y_t)e^t)}}{1 - e^{-2t}} - N(z)\right\|^2\right] \leq \frac{1}{n}\mathbb{E}\left[\frac{\|Y_t\|^2 e^{-2V((z-Y_t)e^t)}}{(1 - e^{-2t})^2}\right].$$

Taking in the trace in the log hessian identity in Proposition 11 yields

$$\mathbb{E}\left[\frac{\|Y_t\|^2 e^{-2V((z-Y_t)e^t)}}{(1 - e^{-2t})^2}\right] = \left(\Delta \log(p_t^{2V}) - \frac{\Delta \log(\pi)}{1 - e^{-2t}} + \|\nabla \log(p_t^{2V})\|^2\right)p_t^{2V} Z_{2V} e^{-td}.$$

Similarly, we have

$$\hat{D} - D = \frac{1}{n}\sum_{i=1}^n e^{-V((z-y_i)e^t)} - D,$$

hence we get using again Proposition 11,

$$\mathbb{E}[(\hat{D} - D)^2] = \frac{1}{n}\left(\mathbb{E}[e^{-2V((z-Y_t)e^t)}] - \mathbb{E}[e^{-2((z-Y_t)e^t)}]^2\right) \leq \frac{p_t^{2V}(z) Z_{2V} e^{-td}}{n}. \quad (26)$$

$\square$

We can now prove Proposition 4.

*Proof.* Define the event $A = (\hat{D} \geq \eta D) \cap (\|\hat{N}\| \leq \kappa \|N\|)$ where $\eta \leq 1, \kappa \geq 1$ are positive scalars to be chosen later. We start to decompose the quadratic error as:

$$\mathbb{E}\left[\left\|\frac{\hat{N}}{\hat{D}} - \frac{N}{D}\right\|^2\right] = \mathbb{E}\left[\left\|\frac{\hat{N}}{\hat{D}} - \frac{N}{D}\right\|^2 \mathbb{1}_A\right] + \mathbb{E}\left[\left\|\frac{\hat{N}}{\hat{D}} - \frac{N}{D}\right\|^2 \mathbb{1}_{\bar{A}}\right].$$

We now separately analyze the first and the second term. For the first term, define

$$\begin{cases} \theta : \mathbb{R}^d \times \mathbb{R}^* \to \mathbb{R} \\ (x,p) \mapsto \left\| \frac{x}{p} - \frac{N}{D} \right\|^2 \end{cases}.$$

The gradient and Hessian of $\theta$ are given by

$$\begin{cases} \nabla\theta(x,p) = \frac{-2}{p}\left( \frac{N}{D} - \frac{x}{p}, \left\| \frac{x}{p} \right\|^2 - \left\langle \frac{x}{p}, \frac{N}{D} \right\rangle \right), \\ \nabla^2\theta(x,p) = \frac{-2}{p^2}\begin{pmatrix} -I_d & (\frac{2x}{p} - \frac{N}{D})^\top \\ (\frac{2x}{p} - \frac{N}{D}) & -3\left\| \frac{x}{p} \right\|^2 + 2\langle \frac{x}{p}, \frac{N}{D}\rangle \end{pmatrix}. \end{cases}$$

We thus make a Taylor expansion of order 2 of $\theta(\hat{N}, \hat{D})$ around $(N, D)$: there exists (a random) $\hat{t} \in [0,1]$ such that

$$\theta(\hat{N}, \hat{D}) = \theta(N, D) + \nabla\theta(N, D)^\top(\hat{N} - N, \hat{D} - D)$$
$$+ \frac{1}{2}(\hat{N} - N, \hat{D} - D)^\top \nabla^2\theta(\hat{N}_{\hat{t}}, \hat{D}_{\hat{t}})(\hat{N} - N, \hat{D} - D).$$

where we denoted $\hat{N}_{\hat{t}} = \hat{t}\hat{N} + (1 - \hat{t})N$ and $\hat{D}_{\hat{t}} = \hat{t}\hat{D} + (1 - \hat{t})D$. The two first terms in the expansion are null and we are left with

$$\theta(\hat{N}, \hat{D}) = \frac{1}{\hat{D}_{\hat{t}}^2}\left( \|\hat{N} - N\|^2 - 2(\langle 2\frac{\hat{N}_{\hat{t}}}{\hat{D}_{\hat{t}}} - \frac{N}{D}, \hat{N} - N\rangle(\hat{D} - D)) \right.$$
$$\left. + \left( 3\frac{\|\hat{N}_{\hat{t}}\|^2}{\hat{D}_{\hat{t}}^2} - 2\langle \frac{\hat{N}_{\hat{t}}}{\hat{D}_{\hat{t}}}, \frac{N}{D}\rangle \right)(\hat{D} - D)^2 \right)$$
$$\leq \frac{1}{\hat{D}_{\hat{t}}^2}\left( \|\hat{N} - N\|^2 + 2\|2\frac{\hat{N}_{\hat{t}}}{\hat{D}_{\hat{t}}} - \frac{N}{D}\|\|\hat{N} - N\||\hat{D} - D| + \left( 3\frac{\|\hat{N}_{\hat{t}}\|^2}{\hat{D}_{\hat{t}}^2} + 2\left\| \frac{\hat{N}_{\hat{t}}}{\hat{D}_{\hat{t}}} \right\|\left\| \frac{N}{D} \right\| \right)(\hat{D} - D)^2 \right).$$

Hence, almost surely over $A$

$$\left\| \frac{\hat{N}}{\hat{D}} - \frac{N}{D} \right\|^2 \leq \frac{1}{\eta^2 D^2}\left( \|\hat{N} - N\|^2 + \frac{6\kappa}{\eta}\left\| \frac{N}{D} \right\|\|\hat{N} - N\||\hat{D} - D| + 5\left\| \frac{N}{D} \right\|^2 \left( \frac{\kappa}{\eta} \right)^2 (\hat{D} - D)^2 \right).$$

Hence, after taking the expectation and applying Cauchy-Schwarz, we obtain

$$\mathbb{E}\left[ \left\| \frac{\hat{N}}{\hat{D}} - \frac{N}{D} \right\|^2 \mathbb{1}_A \right] \leq \frac{1}{\eta^2 D^2}(\mathbb{E}[\|\hat{N} - N\|^2] + \frac{6\kappa}{\eta}\left\| \frac{N}{D} \right\|\mathbb{E}[\|\hat{N} - N\|^2]^{1/2}\mathbb{E}[(\hat{D} - D)^2]^{1/2}$$
$$+ 5\left\| \frac{N}{D} \right\|^2 \left( \frac{\kappa}{\eta} \right)^2 \mathbb{E}[(\hat{D} - D)^2]).$$

Now recall that $D = p_t^V Z_V e^{-td}$ and that $\frac{N}{D} = \nabla\log(p_t^V)$ which, combined with Proposition 12 yields for the first term:

$$\frac{\mathbb{E}[\|\hat{N} - N\|^2]}{\eta^2 D^2} \leq \frac{p_t^{2V} Z_{2V} e^{td}}{\eta^2(p_t^V)^2(Z_V)^2 n}\left( \Delta\log(p_t^{2V}) - \frac{\Delta\log(\pi)}{1 - e^{-2t}} + \|\nabla\log(p_t^{2V})\|^2 \right),$$

for the second term:

$$\frac{6\kappa}{\eta^3 D^2}\left\| \frac{N}{D} \right\|\|\hat{N} - N\||\hat{D} - D| \leq \frac{6\kappa p_t^{2V} Z_{2V} e^{td}}{\eta^3(p_t^V)^2(Z_V)^2 n}\|\nabla\log(p_t^V)\|\left( \Delta\log(p_t^{2V}) - \frac{\Delta\log(\pi)}{1 - e^{-2t}} + \|\nabla\log(p_t^{2V})\|^2 \right)^{1/2}$$

and for the last term:

$$5\left\| \frac{N}{D} \right\|^2 \left( \frac{\kappa}{\eta^2 D} \right)^2 (\hat{D} - D)^2 \leq \frac{5\kappa^2 p_t^{2V} Z_{2V} e^{td}}{\eta^4(p_t^V)^2(Z_V)^2 n}\|\nabla\log(p_t^V)\|^2.$$

Hence we finally obtain

$$\mathbb{E}\left[\left\|\frac{\hat{N}}{\hat{D}}-\frac{N}{D}\right\|^2 \mathbb{1}_A\right] \leq \frac{p_t^{2V} Z_{2V} e^{td}}{n\eta^2 (p_t^V Z_V)^2}\left(\frac{6\kappa}{\eta}\|\nabla\log(p_t^V)\|\left[\Delta\log(p_t^{2V})-\frac{\Delta\log(\pi)}{1-e^{-2t}}+\|\nabla\log(p_t^{2V})\|^2\right]^{1/2}\right.$$

$$\left.+\Delta\log(p_t^{2V})-\frac{\Delta\log(\pi)}{1-e^{-2t}}+6\left(\frac{\kappa}{\eta}\right)^2\|\nabla\log(p_t^V)\|^2\right).$$

$$(27)$$

Let us now handle the quadratic error of the estimator on the complementary $\bar{A}$. We have, using Young's inequality $\|a-b\|^2 \leq 2(\|a\|^2+\|b\|^2)$,

$$\mathbb{E}\left[\left\|\frac{\hat{N}}{\hat{D}}-\frac{N}{D}\right\|^2 \mathbb{1}_{\bar{A}}\right] \leq 2\left\|\frac{N}{D}\right\|^2 \mathbb{P}(\bar{A}) + 2\mathbb{E}\left[\mathbb{1}_{\bar{A}}\frac{\max_i\|y_i\|^2}{(1-e^{-2t})^2}\right].$$

Now, recall that $X_i = \|y_i\|^2(1-e^{-2t})^{-1}$ are $n$ independent variables such that for all $i$, $X_i \sim \chi^2(d)$. Using Hölder inequality for some $p \geq 1$, the second term can be upper-bounded as

$$\mathbb{E}\left[\mathbb{1}_{\bar{A}}\frac{\max_i\|y_i\|^2}{(1-e^{-2t})^2}\right] \leq \frac{1}{1-e^{-2t}}\mathbb{E}[\max_i X_i^p]^{1/p}\mathbb{P}(\bar{A})^{1-1/p}$$

$$\leq \frac{1}{1-e^{-2t}}(n\mathbb{E}[X_1^p])^{1/p}\mathbb{P}(\bar{A})^{1-1/p} \leq \frac{1}{1-e^{-2t}}n^{1/p}(d+2p)\mathbb{P}(\bar{A})^{1-1/p},$$

where we used in the penultimate inequality that the max is smaller than the sum, and in the last one that $\mathbb{E}[X_1^p] = \prod_{i=0}^{p-1}(d+2i)$ when $X_1 \sim \chi^2(d)$ combined with the fact that the geometric mean is lower than the arithmetic mean.

We now upper bound the probability of the event $\bar{A} = (\hat{D} < \eta D) \cup (\|\hat{N}\| > \kappa\|N\|)$. By Chebyshev's inequality, using $\eta < 1$, it holds that

$$\mathbb{P}(\hat{D} < \eta D) \leq \frac{\mathbb{E}[(\hat{D}-D)^2]}{D^2(\eta-1)^2} \leq \frac{p_t^{2V} Z_{2V} e^{td}}{n(p_t^V Z_V)^2(\eta-1)^2} := \frac{U}{n(\eta-1)^2}.$$

Similarly, recalling that $\|N\| = D\|\nabla\log(p_t^V)\|$, we have

$$\mathbb{P}(\|\hat{N}\| > \kappa\|N\|) \leq \frac{\mathbb{E}[\|\hat{N}-N\|^2]}{\|N\|^2(\kappa-1)^2}$$

$$\leq \frac{U}{n\|\nabla\log(p_t^V)\|^2(\kappa-1)^2}\left(\Delta\log(p_t^{2V})-\frac{\Delta\log(\pi)}{1-e^{-2t}}+\|\nabla\log(p_t^V)\|^2\right).$$

We now make a disjunction of cases: if $\|\nabla\log(p_t^V)\| \geq 1$, we pick $\eta = 1/2$ and $\kappa = 3/2$ so we recover

$$\mathbb{E}\left[\left\|\frac{\hat{N}}{\hat{D}}-\frac{N}{D}\right\|^2 \mathbb{1}_{\bar{A}}\right] \leq \frac{8U}{n}\left[\Delta\log(p_t^{2V})-\frac{\Delta\log(\pi)}{1-e^{-2t}}+2\|\nabla\log(p_t^V)\|^2\right]$$

$$+\frac{n^{1/p}(d+2p)}{1-e^{-2t}}\left[\frac{8U}{n}\left(\Delta\log(p_t^{2V})-\frac{\Delta\log(\pi)}{1-e^{-2t}}+\|\nabla\log(p_t^V)\|^2+1\right)\right]^{1-1/p}$$

$$\leq \frac{8U}{n}\left[\Delta\log(p_t^{2V})-\frac{\Delta\log(\pi)}{1-e^{-2t}}+2\|\nabla\log(p_t^V)\|^2\right]$$

$$+\frac{n^{1/p}(d+2p)}{1-e^{-2t}}\left[\frac{8U}{n}\left(\Delta\log(p_t^{2V})-\frac{\Delta\log(\pi)}{1-e^{-2t}}+\|\nabla\log(p_t^V)\|^2+1\right)+1\right].$$

We thus pick $p = \log(n)$ to get

$$\mathbb{E}\left[\left\|\frac{\hat{N}}{\hat{D}}-\frac{N}{D}\right\|^2 \mathbb{1}_{\bar{A}}\right] \leq \frac{16e^2(d+2\log(n))}{n(1-e^{-2t})}\left[U\left(\Delta\log(p_t^{2V})-\frac{\Delta\log(\pi)}{1-e^{-2t}}+\|\nabla\log(p_t^V)\|^2+1\right)+1\right].$$

Combining this with the bound Eq. 27 eventually yields

$$\delta^2 \leq \frac{32e^2(d+2\log(n))}{n(1-e^{-2t})}\left[U\left(\Delta\log(p_t^{2V}) - \frac{\Delta\log(\pi)}{1-e^{-2t}} + \|\nabla\log(p_t^V)\|^2 + \|\nabla\log(p_t^{2V})\|^2 + 1\right) + 1\right],$$

where we used the inequality

$$\|\nabla\log(p_t^V)\|\left[\Delta\log(p_t^{2V}) - \frac{\Delta\log(\pi)}{1-e^{-2t}} + \|\nabla\log(p_t^{2V})\|^2\right]^{1/2}$$

$$\leq \frac{1}{2}\left(\|\nabla\log(p_t^V)\|^2 + \Delta\log(p_t^{2V}) - \frac{\Delta\log(\pi)}{1-e^{-2t}} + \|\nabla\log(p_t^{2V})\|^2\right).$$

In the case where $\|\nabla\log(p_t)\|^2 < 1$, we instead pick $\eta = 1/2$ and $\kappa = 1 + \frac{1}{2\|\nabla\log(p_t^V)\|}$. We obtain that

$$\mathbb{P}(\bar{A}) \leq \frac{4U}{n}\left(\Delta\log(p_t^{2V}) - \frac{\Delta\log(\pi)}{1-e^{-2t}} + \|\nabla\log(p_t^V)\|^2 + 1\right)$$

and as previously, for $p = \log(n)$ we get

$$\mathbb{E}\left[\left\|\frac{\hat{N}}{\hat{D}} - \frac{N}{D}\right\|^2 \mathbb{1}_{\bar{A}}\right] \leq \frac{16e^2(d+2\log(n))}{n(1-e^{-2t})}\left[U\left(\Delta\log(p_t^{2V}) - \frac{\Delta\log(\pi)}{1-e^{-2t}} + \|\nabla\log(p_t^V)\|^2 + 1\right) + 1\right].$$

For this choice of $\kappa, \eta$, the bound on $A$ becomes

$$\mathbb{E}\left[\left\|\frac{\hat{N}}{\hat{D}} - \frac{N}{D}\right\|^2 \mathbb{1}_A\right] \leq \frac{4U}{n}\left(6\left[\Delta\log(p_t^{2V}) - \frac{\Delta\log(\pi)}{1-e^{-2t}} + \|\nabla\log(p_t^{2V})\|^2\right]^{1/2} + \Delta\log(p_t^{2V}) - \frac{\Delta\log(\pi)}{1-e^{-2t}} + 6\right)$$

$$\leq \frac{4U}{n}\left(7(\Delta\log(p_t^{2V}) - \frac{\Delta\log(\pi)}{1-e^{-2t}}) + 6(\|\nabla\log(p_t^{2V})\|^2 + 1)\right).$$

Thus, we obtain as previously

$$\delta^2 \leq \frac{32e^2(d+2\log(n))}{n(1-e^{-2t})}\left[U\left(\Delta\log(p_t^{2V}) - \frac{\Delta\log(\pi)}{1-e^{-2t}} + \|\nabla\log(p_t^V)\|^2 + \|\nabla\log(p_t^{2V})\|^2 + 1\right) + 1\right],$$

$\square$

## D Proof of Proposition 6 and Lemma 8

Before starting the proofs, we recall the following usefull lemma that bounds the second order moment of dissipative distributions.

**Lemma 13** *Let $V$ be such that $\langle\nabla V(x), x\rangle \geq a\|x\|^2 - b$. Then for $\mu \propto e^{-V}$, denoting $m_2$ the second moment of $\mu$, it holds that $m_2 \leq \frac{b+d}{a}$.*

*Proof.* Define the Laplacian of $\mu$ as $L(f) = \Delta f - \langle\nabla V, \nabla f\rangle$ for $f$ sufficiently smooth. By integration by parts, it holds that

$$\int L(f)(x)\mathrm{d}\mu(x) = 0.$$

In particular, for $f(x) = \|x\|^2$, we recover $\int\langle\nabla V(x), \nabla f(x)\rangle\mathrm{d}\mu(x) = 2d$. For $V$ dissipative, it implies

$$a\int\|x\|^2\mathrm{d}\mu(x) \leq 2d + b,$$

or equivalently $m_2 \leq (b+2d)/a$. $\square$

## D.1 Proof of Lemma 8

*Proof.* The first inequality was shown in the Lemma above. For the Fisher information, it holds that

$$\mathcal{I}(\mu, \pi) = \int \|\nabla V(x) - x\|^2 \mathrm{d}\mu(x),$$

$$\leq 2 \int \langle \nabla V(x), \nabla V(x) e^{-V(x)} \rangle / Z_V \mathrm{d}x + 2m_2,$$

$$= 2 \int \Delta V \mathrm{d}\mu(x) + 2m_2,$$

$$\leq 2\beta d + 2m_2.$$

There remains to lower-bound $\mu(0)$. Denote $x^*$ a global minimizer of $V$. By dissipativity, it must hold that $\|x^*\|^2 \leq b/a$. Now observe that

$$V(x^*) - V(0) = \int_0^1 (x^*)^\top \nabla^2 V(tx^*) x^* \mathrm{d}t \leq \beta \|x^*\|^2.$$

Combined with the fact that for all $x \in \mathbb{R}^d$, it holds that $V(x) - V(x^*) \geq 0$, be recover that $V(x) - V(0) \geq -\beta \|x^*\|^2 \geq -\beta b/a$. Furthermore, for $0 < \delta < 1$, we have

$$V(x) - V(0) = \int_0^1 \langle \nabla V(tx), x \rangle \mathrm{d}t,$$

$$= \int_\delta^1 \langle \nabla V(tx), x \rangle \mathrm{d}t + V(\delta x) - V(0),$$

$$\geq \int_\delta^1 \langle \nabla V(tx), tx \rangle / t \mathrm{d}t - \beta b/a,$$

$$\geq \int_\delta^1 at\|x\|^2 - b/t \mathrm{d}t - \beta b/a,$$

$$= a\|x\|^2/2 - a\delta^2 \|x\|^2/2 + b\log(\delta) - \beta b/a.$$

In particular, for $\delta = 1/\sqrt{2}$, we obtain

$$V(x) - V(0) \geq a\|x\|^2/4 - b(\beta/a + \log(2)/2).$$

Hence, we have

$$\mu(0) = \frac{e^{-V(0)}}{\int e^{-V(x)} \mathrm{d}x},$$

$$= \frac{1}{\int e^{-(V(x) - V(0))} \mathrm{d}x},$$

$$\leq \frac{1}{\int e^{-a\|x\|^2/4 + b(\beta/a + \log(2)/2)} \mathrm{d}x}$$

$$= e^{-b(\beta/a + \log(2)/2)} (a/2)^{d/2} (2\pi)^{-d/2}.$$

Since $\beta/a \geq 1$ and $\log(2)/2 \leq 1$, we recover that $\log\left(\mu(0)^{-2/d}\right) \leq 4\beta b/ad + 2\pi + \log(2/a)$.

$\square$

## D.2 Proof of Proposition 6

Recall that the intermediate scores read

$$\nabla \log\left(p_t^V\right)(z) = \frac{z - e^{-t}\mathbb{E}_{q_{t,z}}[Y]}{1 - e^{-2t}},$$

with $q_{t,z}(x) \propto e^{-V(x)} e^{-\frac{\|e^{-t}x - z\|^2}{2(1 - e^{-2t})}}$. In particular, if $V$ is dissipative with constant $a, b$ then it holds that

$$\langle -\nabla \log(q_{t,z})(x), x \rangle = \langle \nabla V(x) + \frac{e^{-t}(e^{-t}x - z)}{1 - e^{-2t}}, x \rangle,$$

$$\geq a\|x\|^2 - b + \frac{e^{-t}}{1 - e^{-2t}}(e^{-t}\|x\|^2 - \langle z, x \rangle).$$

Now recall that $e^{-t}\|x\|^2 - \langle z, x \rangle \geq -e^t\|z\|^2/4$ which yields

$$\langle -\nabla \log(q_{t,z})(x), x \rangle \geq a\|x\|^2 - (b + \|z\|^2/(4(1 - e^{-2t}))).$$

Hence, using Lemma 13, it holds that

$$\mathbb{E}_{q_{t,z}}[\|Y\|^2] \leq \frac{2b + \|z\|^2/(2(1 - e^{-2t})) + d}{a}.$$

Hence we recover that

$$\begin{aligned}
\|\nabla \log(p_t^V)(z)\|^2 &\leq \frac{2\|z\|^2}{(1 - e^{-2t})^2} + \frac{2e^{-2t}\mathbb{E}_{q_{t,z}}[\|Y\|^2]}{(1 - e^{-2t})^2}, \\
&\leq \frac{2\|z\|^2}{(1 - e^{-2t})^2} + \frac{e^{-2t}}{(1 - e^{-2t})^2}\left(\frac{2(2b + d)}{a} + \frac{\|z\|^2}{(1 - e^{-2t})a}\right), \\
&= \frac{\|z\|^2}{(1 - e^{-2t})^2}\left(2 + \frac{e^{-2t}}{a(1 - e^{-2t})}\right) + \frac{2e^{-2t}(2b + d)}{a(1 - e^{-2t})^2}.
\end{aligned}$$

Similarly, recall that

$$\nabla^2 \log(p_t^V)(z) - \nabla^2 \log(\pi)(z) = \frac{e^{-2t}}{1 - e^{-2t}}\left(\frac{\mathrm{Cov}_{q_{t,z}}(X)}{1 - e^{-2t}} - I_d\right),$$

from which we can deduce $\Delta \log(p_t^{2V}) \leq \frac{e^{-2t}}{(1 - e^{-2t})^2}\mathbb{E}_{q_{t,z}}[\|Y\|^2]$ which yields again

$$\Delta \log(p_t^{2V}) \leq \frac{e^{-2t}}{(1 - e^{-2t})^2}\left(\frac{\|z\|^2}{2a(1 - e^{-2t})} + \frac{2b + d}{a}\right).$$

# E   Proof of Lemma 7

Before starting the proof, we recall the result of Mikulincer and Shenfeld [2023].

**Proposition 14** *Let $\mu$ be a $\beta$-semi-log-convex probability distribution. Then, denoting $p_t^V$ the distribution of the forward process in Eq. 1 and $\pi$ the density of the standard Gaussian, it holds for all $z \in \mathbb{R}^d$ that*

$$\nabla^2 \log(\pi)(z) - \nabla^2 \log(p_t^V)(z) \preceq \frac{(\beta - 1)e^{-2t}}{(1 - e^{-2t})(\beta - 1) + 1}I_d. \tag{28}$$

*Proof.* Define the Ornstein-Uhlenbeck semi-group $Q_t$ as

$$\begin{aligned}
Q_t(g)(z) &= \int g(ze^{-t} + \sqrt{1 - e^{-2t}}y)e^{-\frac{\|y\|^2}{2}}(2\pi)^{-d/2}\mathrm{d}y \\
&= \frac{1}{\sqrt{1 - e^{-2t}}}\int g(u)e^{-\frac{\|u - e^{-t}z\|^2}{2(1 - e^{-2t})}}(2\pi)^{-d/2}\mathrm{d}u,
\end{aligned}$$

for all function $g$ integrable w.r.t. the standard Gaussian measure. Taking $g$ as $f = \frac{\mathrm{d}\mu}{\mathrm{d}\pi}$ with $\pi$ the standard Gaussian, we obtain that

$$\begin{aligned}
Q_t(f)(z) &= \frac{1}{Z_V\sqrt{1 - e^{-2t}}}\int e^{-V(u)}e^{\frac{\|u\|^2}{2}}e^{-\frac{\|u - e^{-t}z\|^2}{2(1 - e^{-2t})}}\mathrm{d}u \\
&= \frac{1}{Z_V\sqrt{1 - e^{-2t}}}\int e^{-V(u)}e^{\frac{\|u\|^2(1 - e^{-2t}) - \|u\|^2 + \langle z, ue^{-t}\rangle - e^{-2t}\|z\|^2}{2(1 - e^{-2t})}}\mathrm{d}u \\
&= \frac{1}{Z_V\sqrt{1 - e^{-2t}}}\int e^{-V(u)}e^{\frac{-\|ue^{-t} - z\|^2 + \|z\|^2 - e^{-2t}\|z\|^2}{2(1 - e^{-2t})}}\mathrm{d}u \\
&= \frac{e^{\frac{\|z\|^2}{2}}}{Z_V\sqrt{1 - e^{-2t}}}\int e^{-V(u)}e^{-\frac{\|ue^{-t} - z\|^2}{2(1 - e^{-2t})}}\mathrm{d}u.
\end{aligned}$$

In particular, we remark that $\nabla \log(Q_t(f)) = \nabla \log(p_t^V) - \nabla \log(\pi)$. Now, the quantity $\nabla \log(Q_t(f))$ was studied in Mikulincer and Shenfeld [2023] and they prove in Lemma 5 that for all $z$

$$\nabla^2 \log(Q_t(f))(z) \succeq \frac{(1 - \beta)e^{-2t}}{(1 - e^{-2t})(\beta - 1) + 1}I_d,$$

which is equivalent to

$$\nabla^2 \log(\pi)(z) - \nabla^2 \log\!\big(p_t^V\big)(z) \preceq \frac{(\beta-1)e^{-2t}}{(1-e^{-2t})(\beta-1)+1} I_d \,.$$

$\square$

Before proving Lemma 7, we introduce this preliminary result on the evolution of $\Phi_t$.

**Lemma 15** (Evolution of the ratio) *Let $t > 0$, it holds that*

$$\partial_t \Phi_t = \Phi_t \left( \Delta \log(\Phi_t) - \langle \nabla \log(\Phi_t), \nabla \log(\pi) \rangle + \|\nabla \log(\Phi_t)\|^2 + 2\langle \nabla \log\!\big(p_t^V\big), \nabla \log(\Phi_t) \rangle \right) \,.$$

*Proof.* Recall that the log-density $\log\!\big(p_t^V\big)$ evolves as

$$\partial_t \log\!\big(p_t^V\big) = \Delta \log\!\big(p_t^V\big) + \|\nabla \log\!\big(p_t^V\big)\|^2 - \langle \nabla \log\!\big(p_t^V\big), \nabla \log(\pi) \rangle - \Delta \log(\pi) \,.$$

Hence, we deduce that $\log(\Phi_t)$ evolves as

$$\partial_t \log(\Phi_t) = \Delta \log(\Phi_t) - \langle \nabla \log(\Phi_t), \nabla \log(\pi) \rangle + \|\nabla \log\!\big(p_t^{2V}\big)\|^2 - \|\nabla \log\!\big(p_t^V\big)\|^2 \,.$$

The difference of quadratic terms can be expressed as

$$\|\nabla \log\!\big(p_t^{2V}\big)\|^2 - \|\nabla \log\!\big(p_t^V\big)\|^2 = \|\nabla \log(\Phi_t) + \nabla \log\!\big(p_t^V\big)\|^2 - \|\nabla \log\!\big(p_t^V\big)\|^2$$
$$= \|\nabla \log(\Phi_t)\|^2 + 2\langle \nabla \log\!\big(p_t^V\big), \nabla \log(\Phi_t) \rangle \,,$$

which allows to recover

$$\partial_t \log(\Phi_t) = \Delta \log(\Phi_t) - \langle \nabla \log(\Phi_t), \nabla \log(\pi) \rangle + \|\nabla \log(\Phi_t)\|^2 + 2\langle \nabla \log\!\big(p_t^V\big), \nabla \log(\Phi_t) \rangle \,.$$

$\square$

We now provide the proof of Lemma 7.

*Proof.* Until the rest of the proof, the dependence on $z$ of the integrand shall be implied unless expressed explicitly. We start by differentiating $m_0(\Phi_t)$ with respect to $t$:

$$\partial_t m_0(\Phi_t) = \int \partial_t \Phi_t \mathrm{d}z$$
$$= \int \Phi_t \left( \Delta \log(\Phi_t) - \langle \nabla \log(\Phi_t), \nabla \log(\pi) \rangle + \|\nabla \log(\Phi_t)\|^2 + 2\langle \nabla \log\!\big(p_t^V\big), \nabla \log(\Phi_t) \rangle \right) \mathrm{d}z \,,$$

where we used Lemma 15 to compute $\partial_t \Phi_t$. Using integration by parts, the first term reads

$$\int \Phi_t \Delta \log(\Phi_t) \mathrm{d}z = -\int \langle \nabla \Phi_t, \nabla \log(\Phi_t) \rangle \mathrm{d}z = -\int \Phi_t \|\nabla \log(\Phi_t)\|^2 \mathrm{d}z,$$

hence the first and the third terms cancel and we recover

$$\partial_t m_0(\Phi_t) = 2\int \langle \nabla \log\!\big(p_t^V\big), \nabla \Phi_t \rangle \mathrm{d}z - \int \langle \nabla \log(\pi), \nabla \Phi_t \rangle \mathrm{d}z \,.$$

Using integration by parts again, we recover

$$\partial_t m_0(\Phi_t) = \int \Delta \log(\pi) \Phi_t \mathrm{d}z - 2\int \Delta \log\!\big(p_t^V\big) \Phi_t \mathrm{d}z$$
$$= 2\int (\Delta \log(\pi) - \Delta \log\!\big(p_t^V\big)) \Phi_t \mathrm{d}z - \int \Delta \log(\pi) \Phi_t \mathrm{d}z$$
$$= dm_0(\Phi_t) + 2\int (\Delta \log(\pi) - \Delta \log\!\big(p_t^V\big)) \Phi_t \mathrm{d}z \,.$$

Using Proposition 14, since $\mu$ is $\beta$-semi-log-convex, the term $(\Delta \log(\pi) - \Delta \log\!\big(p_t^V\big))$ can be upper-bounded uniformly by $\frac{d(\beta-1)e^{-2t}}{(1-e^{-2t})(\beta-1)+1}$ so we eventually get

$$\partial_t m_0(\Phi_t) \le dm_0(\Phi_t) \left( 1 + \frac{2(\beta-1)e^{-2t}}{(1-e^{-2t})(\beta-1)+1} \right) \,.$$

Hence we can use Gronwall's lemma which yields

$$m_0(\Phi_t) \leq m_0(\Phi_0) \exp\left(d \int_0^t \left(1 + \frac{2(\beta-1)e^{-2s}}{(1-e^{-2s})(\beta-1)+1}\right) ds\right).$$

Denoting by $Z_V$ (resp. $Z_{2V}$) the normalizing constant of $e^{-V}$ (resp. $e^{-2V}$), the term $m_0(\Phi_0)$ reads

$$m_0(\Phi_0) = \int \frac{p_0^{2V}(z)}{p_0^V(z)} dz = \frac{Z_V}{Z_{2V}} \int \frac{e^{-2V(z)}}{e^{-V(z)}} dz = \frac{(Z_V)^2}{Z_{2V}}.$$

Finally, let us compute the integral above. Making the change of variable $u = e^{-2s}(\beta-1)$ we have $du = -2(\beta-1)e^{-2s}ds$ which yields

$$\int_0^t \frac{2(\beta-1)e^{-2s}}{(1-e^{-2s})(\beta-1)+1} ds = -\int_{\beta-1}^{(\beta-1)e^{-2t}} \frac{1}{\beta-u} du$$

$$= [\log(\beta-u)]_{\beta-1}^{(\beta-1)e^{-2t}}$$

$$= \log(\beta - (\beta-1)e^{-2t})$$

$$= \log(\beta(1-e^{-2t}) + e^{-2t}).$$

Hence we recover

$$m_0(\Phi_t) \leq \frac{e^{td}(Z_V)^2}{Z_{2V}} (\beta(1-e^{-2t}) + e^{-2t})^d. \qquad \square$$

In order to recover a bound on the second moment of $\Phi_t$, we need several intermediate results. We first prove that the maximum of the ratio decreases through time.

**Lemma 16** (Decrease of the maximum of $\Phi$) *The maximum of the ratio $\Phi_t$ decreases with $t$.*

*Proof.* Let $z_t$ be a point where $\Phi_t$ attains its maximum and denote $M_t = \log(\Phi_t)(z_t)$. By the implicit function theorem, $z_t$ is differentiable hence we can compute $\partial_t M_t$ as

$$\partial_t M_t = \partial_t \log(\Phi_t)(z_t) + \langle \partial_t z_t, \nabla \log(\Phi_t)(z_t) \rangle$$
$$= \Delta \log(\Phi_t)(z_t).$$

Since $z_t$ is a maximum, we have in particular $\Delta \log \Phi_t(z_t) \leq 0$ which implies that $M_t$ decreases. $\square$

We then derive an upper-bound on the maximum of a log-smooth distribution.

**Proposition 17** (Upper-bound of the maximum) *If $\mu$ is $\beta$-semi-log-convex then it holds that $\frac{d\mu}{dz} \leq \left(\frac{\beta}{2\pi}\right)^{\frac{d}{2}}$.*

*Proof.* Recall that the density of $\mu$ can be re-written as

$$\frac{d\mu}{dz} = \frac{e^{-(V(z)-V_*)}}{\int_z e^{-(V(z)-V_*)}dz},$$

where $V_*$ the minimum of $V$ attained for some $z_*$. By definition $e^{-(V(z)-V_*)} \leq 1$ for all $z$. Furthermore, since $V$ verifies $\nabla^2 V \preceq \beta I_d$, we are ensured that

$$V(z) - V_* \leq \beta \frac{\|z-z_*\|^2}{2},$$

which implies in particular that

$$\frac{1}{\int_z e^{-(V(z)-V_*)}dz} \leq \left(\frac{\beta}{2\pi}\right)^{\frac{d}{2}}. \qquad \square$$

Using the previous result, we can derive an upper-bound on the integrated squared gradient at $0$.

**Lemma 18** (Upper-bound integrated gradient) *Let $\mu \propto e^{-V}$ be a $\beta$-semi-log-convex measure. Denoting $\mu(0)$ the density of $\mu$ with respect to the Lebesgue measure at $0$, it holds that*

$$\int_0^t \|\nabla \log(p_s^V)\|^2(0)\mathrm{d}s \leq -\log(\mu(0)) + \frac{d}{2}\log\left(\frac{\beta}{2\pi}\right).$$

*Proof.* Denoting $\pi$ the density of the standard $d$ dimensional Gaussian, recall that the density $p_t^V$ evolves as

$$\partial_t p_t^V = \nabla \cdot \left(p_t^V \nabla \log\left(\frac{p_t^V}{\pi}\right)\right),$$

which can also be re-written as

$$\begin{aligned}
\partial_t \log(p_t^V) &= \frac{\Delta p_t^V}{p_t^V} - \langle \nabla \log(p_t^V), \nabla \log(\pi)\rangle - \Delta \log(\pi) \\
&= \Delta \log(p_t^V) + \|\nabla \log(p_t^V)\|^2 - \langle \nabla \log(p_t^V), \nabla \log(\pi)\rangle - \Delta \log(\pi).
\end{aligned}$$

In particular, for $z = 0$ this yields

$$\partial_t \log(p_t^V)(0) = \Delta \log(p_t^V)(0) + \|\nabla \log(p_t^V)\|^2(0) - \Delta \log(\pi)(0),$$

which implies

$$\int_0^t \|\nabla \log(p_s^V)\|^2(0)\mathrm{d}s = \log(p_t^V)(0) - \log(p_0^V)(0) + \int_0^t \Delta \log(\pi)(0) - \Delta \log(p_t^V)(0)\mathrm{d}s.$$

Using the uniform upper-bound of Proposition 14, the second term can upper-bounded as

$$\begin{aligned}
\int_0^t \Delta \log(\pi)(0) - \Delta \log(p_t^V)(0)\mathrm{d}s &\leq d\int_0^t \frac{(\beta-1)e^{-2s}}{(1-e^{-2s})(\beta-1)+1}\mathrm{d}s \\
&= \frac{d}{2}(\beta-1)\int_{e^{-2t}}^1 \frac{1}{\beta - u(\beta-1)}\mathrm{d}u \\
&= \frac{d}{2}\log\left(\beta(1-e^{-2t}) + e^{-2t}\right).
\end{aligned}$$

Furthermore, Proposition 14 shows that $-\nabla^2 \log(p_t^V) \preceq \frac{\beta}{\beta(1-e^{-2t})+e^{-2t}}$. Thus, using Proposition 17, we recover that $\log(p_t^V)(0) \leq \frac{d}{2}\log\left(\frac{\beta}{\beta(1-e^{-2t})+e^{-2t}}\right) - \frac{d}{2}\log(2\pi)$. In particular, we recover

$$\int_0^t \|\nabla \log(p_s^V)\|^2(0)\mathrm{d}s \leq -\log\left(\frac{\mathrm{d}\mu}{\mathrm{d}x}(0)\right) + \frac{d}{2}\log(\beta) - \frac{d}{2}\log(2\pi). \quad \square$$

We can now bound the first order moment of $\Phi_t$

**Lemma 19** *Let $\mu$ be a $\beta$-semi-log-convex measure with finite second moment $m_2$. It holds that*

$$m_1(\Phi_t) \leq \frac{(Z_V)^2}{Z_{2V}}e^{t(d+1)}(\beta(1-e^{-2t})+e^{-2t})^d\left(\sqrt{m_2} + \sqrt{-2\log(\mu(0)) + d\log\left(\frac{\beta}{2\pi}\right)} + 2\sqrt{d\beta}\right).$$

*Proof.* We differentiate $m_1(\Phi_t)$ and we recover

$$\partial_t m_1(\Phi_t) = \int \Phi_t \left(\Delta \log(\Phi_t) - \langle \nabla \log(\Phi_t), \nabla \log(\pi)\rangle + \|\nabla \log(\Phi_t)\|^2 + 2\langle \nabla \log(p_t^V), \nabla \log(\Phi_t)\rangle\right)\|z\|\mathrm{d}z.$$

Integration by parts of the first term yields

$$\int \Phi_t \Delta \log(\Phi_t)\|z\|\mathrm{d}z = -\int \Phi_t \|\nabla \log(\Phi_t)\|^2\|z\| + \langle \nabla\Phi_t, \frac{z}{\|z\|}\rangle\mathrm{d}z,$$

hence the squared gradients terms cancel and we recover

$$\partial_t m_1(\Phi_t) = \int \langle 2\nabla \log(p_t^V) - \nabla \log(\pi), \nabla\Phi_t\rangle\|z\|\mathrm{d}z - \int\langle\nabla\Phi_t, \frac{z}{\|z\|}\rangle\mathrm{d}z.$$

Let us denote by $A$ the first term above. Integration by parts yields:

$$A = \int \Phi_t(\Delta \log(\pi) - 2\Delta \log(p_t^V))\|z\|\mathrm{d}z + \int \Phi_t\langle \nabla \log(\pi) - 2\nabla \log(p_t^V), \frac{z}{\|z\|}\rangle \mathrm{d}z$$

$$= -\int \Phi_t \Delta \log(\pi)\|z\|\mathrm{d}z + 2\int \Phi_t(\Delta \log(\pi) - \Delta \log(p_t^V))\|z\|\mathrm{d}z$$

$$- \int \Phi_t\langle \nabla \log(\pi), \frac{z}{\|z\|}\rangle \mathrm{d}z + 2\int \Phi_t\langle \nabla \log(\pi) - \nabla \log(p_t^V), \frac{z}{\|z\|}\rangle \mathrm{d}z$$

$$= (d+1)m_1(\Phi_t) + 2\int \Phi_t(\Delta \log(\pi) - \Delta \log(p_t^V))\|z\|\mathrm{d}z + 2\int \Phi_t\langle \nabla \log(\pi) - \nabla \log(p_t^V), \frac{z}{\|z\|}\rangle \mathrm{d}z.$$

Using the upper bound given in Proposition 14, we get $2\int \Phi_t(\Delta \log(\pi) - \Delta \log(p_t^V))\|z\|\mathrm{d}z \le$ $2d\frac{(\beta-1)e^{-2t}}{(\beta-1)(1-e^{-2t})+1}m_1(\Phi_t)$. Similarly, we re-write the second term as

$$2\int \Phi_t\langle \nabla \log(\pi) - \nabla \log(p_t^V), \frac{z}{\|z\|}\rangle \mathrm{d}z = 2\int \Phi_t\langle \nabla \log\left(\frac{\pi}{p_t^V}\right)(z) - \nabla \log\left(\frac{\pi}{p_t^V}\right)(0), \frac{z}{\|z\|}\rangle \mathrm{d}z$$

$$+ 2\int \Phi_t\langle \nabla \log\left(\frac{\pi}{p_t^V}\right)(0), \frac{z}{\|z\|}\rangle \mathrm{d}z$$

$$\le 2m_1(\Phi_t)\frac{(\beta-1)e^{-2t}}{(\beta-1)(1-e^{-2t})+1} + 2\|\nabla \log(p_t^V)(0)\|m_0(\Phi_t).$$

Let us now handle the term $B = -\int \langle \nabla \Phi_t, \frac{z}{\|z\|}\rangle \mathrm{d}z$. In one dimension, $B = 2\Phi_t(0) \le \max(\Phi_t)$ and for $d \ge 2$, we have

$$B = \int \frac{\Phi_t(d-1)}{\|z\|}\mathrm{d}z$$

$$= \int_{B_R} \frac{\Phi_t(d-1)}{\|z\|}\mathrm{d}z + \int_{\overline{B_R}} \frac{\Phi_t(d-1)}{\|z\|}\mathrm{d}z$$

$$\le \max(\Phi_t)\int_{B_R} \frac{d-1}{\|z\|}\mathrm{d}z + \frac{d-1}{R}m_0(\Phi_t)$$

$$= \max(\Phi_t)\frac{2\pi^{d/2}}{\Gamma(d/2)}R^{d-1} + \frac{d-1}{R}m_0(\Phi_t).$$

Using Lemma 16 and Proposition 17, we have that $\max(\Phi_t) \le \max(\Phi_0) = \frac{(Z_V)^2}{Z_{2V}}\max(\frac{\mathrm{d}\mu}{\mathrm{d}x}) \le$ $\frac{(Z_V)^2}{Z_{2V}}(\frac{\beta}{2\pi})^{d/2}$. Hence, if we pick $R = \left(Z_{2V}m_0(\Phi_t)\Gamma(d/2)/Z_V^2\right)^{1/d}\beta^{-1/2}2^{1/2-1/d}$, we get as an upper-bound for $B$:

$$B \le \frac{Z_V^2}{Z_{2V}}d\sqrt{\beta}2^{1/d-1/2}(m_0(\Phi_t)Z_{2V}/Z_V^2)^{\frac{d-1}{d}}\Gamma(d/2)^{-1/d}$$

$$\le \frac{2Z_V^2}{Z_{2V}}\sqrt{d\beta}(m_0(\Phi_t)Z_{2V}/Z_V^2)^{\frac{d-1}{d}}.$$

In particular, we recover that

$$\partial_t m_1(\Phi_t) \le (d+1)\left(1 + 2\frac{(\beta-1)e^{-2t}}{(\beta-1)(1-e^{-2t})+1}\right)m_1(\Phi_t)$$

$$+ 2\|\nabla \log(p_t^V)(0)\|m_0(\Phi_t) + \frac{2Z_V^2}{Z_{2V}}\sqrt{\beta d}(m_0(\Phi_t)Z_{2V}/Z_V^2)^{\frac{d-1}{d}}, \quad (29)$$

hence using Gronwall lemma, we have that

$$m_1(\Phi_t) \le e^{t(d+1)}(\beta(1-e^{-2t}) + e^{-2t})^{d+1}m_1(\Phi_0)$$

$$+ 2\int_0^t \left[\|\nabla \log(p_s^V)(0)\|m_1(\Phi_s) + \sqrt{d\beta}(m_0(\Phi_s)Z_{2V}/Z_V^2)^{\frac{d-1}{d}}\right]\frac{e^{t(d+1)}(\beta(1-e^{-2t}) + e^{-2t})^{d+1}}{e^{s(d+1)}(\beta(1-e^{-2s}) + e^{-2s})^{d+1}}\mathrm{d}s.$$

Using Lemma 7, we have that $m_0(\Phi_s) \leq \frac{Z_V^2}{Z_{2V}} e^{sd}(\beta(1 - e^{-2s}) + e^{-2s})^d$ hence the first term of the integral is upper-bounded as:

$$\int_0^t \frac{e^{-s(d+1)} m_0(\Phi_s) \|\nabla \log(p_s^V)(0)\|}{(\beta(1 - e^{-2s}) + e^{-2s})^{d+1}} \mathrm{d}s \leq \frac{Z_V^2}{Z_{2V}} \int_0^t \|\nabla \log(p_s^V)(0)\| \frac{e^{-s}}{\beta(1 - e^{-2s}) + e^{-2s}} \mathrm{d}s \,.$$

By Cauchy-Schwarz it holds that

$$\int_0^t \frac{\|\nabla \log(p_s^V)(0)\| e^{-s}}{L(1 - e^{-2s}) + e^{-2s}} \mathrm{d}s \leq \sqrt{\int_0^t \|\nabla \log(p_s^V)(0)\|^2 \mathrm{d}s} \sqrt{\int_0^t \frac{e^{-2s}}{(\beta(1 - e^{-2s}) + e^{-2s})^2} \mathrm{d}s} \,.$$

The integral term is given by

$$\begin{aligned}
\int_0^t \frac{e^{-2s}}{(\beta(1 - e^{-2s}) + e^{-2s})^2} \mathrm{d}s &= \frac{1}{2} \int_{e^{-2t}}^1 \frac{1}{(\beta(1 - u) + u)^2} \mathrm{d}u \\
&= \frac{1}{2(1 - \beta)} [-\frac{1}{\beta(1 - u) + u}]_{e^{-2t}}^1 \\
&= \frac{1}{2(1 - \beta)} (\frac{1}{\beta(1 - e^{-2t}) + e^{-2t}} - 1) \\
&= \frac{1 - e^{-2t}}{2(\beta(1 - e^{-2t}) + e^{-2t})}
\end{aligned}$$

Similarly,

$$\begin{aligned}
\int_0^t m_0(\Phi_s)^{\frac{d-1}{d}} \frac{e^{-s(d+1)}}{(\beta(1 - e^{-2s}) + e^{-2s})^{d+1}} \mathrm{d}s &\leq \int_0^t \frac{e^{-2s}}{(\beta(1 - e^{-2s}) + e^{-2s})^2} \mathrm{d}s \\
&= \frac{1 - e^{-2t}}{2(\beta(1 - e^{-2t}) + e^{-2t})}
\end{aligned}$$

Hence, we obtain

$$m_1(\Phi_t) \leq \frac{Z_V^2}{Z_{2V}} e^{t(d+1)}(\beta(1 - e^{-2t}) + e^{-2t})^{d+1} \left( \frac{Z_{2V}}{Z_V^2} m_1(\Phi_0) + \sqrt{-2\log(\mu(0)) + d\log\left(\frac{\beta}{2\pi}\right)} + \sqrt{d\beta} \right) \,.$$

Finally, $m_1(\Phi_0) = \frac{(Z_V)^2}{Z_{2V}} \int \|z\| \frac{e^{-V(z)}}{\int e^{-V} \mathrm{d}z} \mathrm{d}z = \frac{(Z_V)^2}{Z_{2V}} \sqrt{m_2(\mu)}.$ $\qquad\square$

We can now derive our upper-bound on $m_2(\Phi_t)$.

*Proof.* We start by differentiating $m_2(\Phi_t)$:

$$\begin{aligned}
\partial_t m_2(\Phi_t) &= \int \|z\|^2 \partial_t \Phi_t \mathrm{d}z \\
&= \int \|z\|^2 (\mathrm{div}(\nabla \Phi_t) - \langle \nabla \Phi_t, \nabla \log(\pi) \rangle + 2\langle \nabla \log(p_t^V), \nabla \Phi_t \rangle) \mathrm{d}z \\
&= -\int 2\langle z, \nabla \Phi_t \rangle \mathrm{d}z + \int (\Delta \log(\pi) - 2\Delta \log(p_t^V)) \|z\|^2 \Phi_t \mathrm{d}z + 2\int \langle \nabla \log(\pi) - 2\log(p_t^V), z \rangle \Phi_t \mathrm{d}z \\
&= -2 \int \Delta \log(\pi) \Phi_t \mathrm{d}z - \int \Delta \log(\pi) \|z\|^2 \Phi_t \mathrm{d}z - 2\int \langle \nabla \log(\pi), z \rangle \Phi_t \mathrm{d}z \\
&\quad + 2\int (\Delta \log(\pi) - \Delta \log(p_t^V)) \|z\|^2 \Phi_t \mathrm{d}z + 4\int \langle \nabla \log(\pi) - \nabla \log(p_t^V), z \rangle \Phi_t \mathrm{d}z \\
&= 2dm_0(\Phi_t) + dm_2(\Phi_t) + 2m_2(\Phi_t) \\
&\quad + 2\int (\Delta \log(\pi) - \Delta \log(p_t^V)) \|z\|^2 \Phi_t \mathrm{d}z + 4\int \langle \nabla \log(\pi) - \nabla \log(p_t^V), z \rangle \Phi_t \mathrm{d}z \,.
\end{aligned}$$

The first term $\int (\Delta \log(\pi) - \Delta \log(p_t^V))\|z\|^2 \Phi_t \mathrm{d}z$ is upper-bounded by $\frac{(\beta-1)e^{-2t}}{(1-e^{-2t})(\beta-1)+1}dm_2(\Phi_t)$ and for the second term we have

$$
\begin{aligned}
\int \langle \nabla \log(\pi) - \nabla \log(p_t^V), z\rangle \Phi_t \mathrm{d}z &= \int \langle \nabla \log\left(\frac{\pi}{p_t^V}\right)(z) - \nabla \log\left(\frac{\pi}{p_t^V}\right)(0), z\rangle \Phi_t \mathrm{d}z \\
&\quad + \int \langle \nabla \log\left(\frac{\pi}{p_t^V}\right)(0), z\rangle \Phi_t \mathrm{d}z \\
&\leq \int \frac{(L-1)e^{-2t}}{(1-e^{-2t})(\beta-1)+1}\|z\|^2 \Phi_t \mathrm{d}z + \|\log(p_t^V)(0)\|\int \|z\|\Phi_t \mathrm{d}z \\
&= \frac{(\beta-1)e^{-2t}}{(1-e^{-2t})(\beta-1)+1}m_2(\Phi_t) + \|\nabla \log(p_t^V)(0)\|m_1(\Phi_t).
\end{aligned}
$$

Hence we recover

$$
\partial_t m_2(\Phi_t) \leq (d+2)\left(1 + \frac{2(\beta-1)e^{-2t}}{(1-e^{-2t})(\beta-1)+1}\right)m_2(\Phi_t) + 4\|\nabla \log(p_t^V)(0)\|m_1(\Phi_t) + 2dm_0(\Phi_t).
$$

We now use the Gronwall lemma to obtain

$$
m_2(\Phi_t) \leq e^{t(d+2)}(\beta(1-e^{-2t})+e^{-2t})^{d+2}\left(m_2(\Phi_0) + 2\int_0^t \frac{e^{-s(d+2)}(2m_1(\Phi_s)\|\nabla \log(p_s^V)(0)\| + dm_0(\Phi_s))}{(\beta(1-e^{-2s})+e^{-2s})^{(d+2)}}\mathrm{d}s\right)
$$

Recalling the upper-bound $m_1(\Phi_t) \leq e^{t(d+1)}(\beta(1-e^{-2t})+e^{-2t})^{d+1}C$ where $C$ is defined in Lemma 19, we upper-bound the integral term as

$$
\begin{aligned}
\int_0^t \frac{e^{-s(d+2)}m_1(\Phi_s)\|\nabla \log(p_s^V)(0)\|}{(\beta(1-e^{-2s})+e^{-2s})^{(d+2)}}\mathrm{d}s &\leq C\int_0^t \frac{e^{-s}}{\beta(1-e^{-2s})+e^{-2s}}\|\nabla \log(p_s^V)(0)\|\mathrm{d}s \\
&\leq C\sqrt{\int_0^t \frac{e^{-2s}}{(\beta(1-e^{-2s})+e^{-2s})^2}\mathrm{d}s}\sqrt{\int_0^t \|\nabla \log(p_t^V)(0)\|^2\mathrm{d}s} \\
&= C\sqrt{\frac{1-e^{-2t}}{2}}\sqrt{\int_0^t \|\nabla \log(p_t^V)(0)\|^2\mathrm{d}s}.
\end{aligned}
$$

Similarly,

$$
\begin{aligned}
\int_0^t \frac{e^{-s(d+2)}}{(\beta(1-e^{-2s})+e^{-2s})^{(d+2)}}m_0(\Phi_s)\mathrm{d}s &\leq \int_0^t \frac{e^{-2s}}{(\beta(1-e^{-2s})+e^{-2s})^2} \\
&= \frac{1-e^{-2t}}{2}.
\end{aligned}
$$

Hence we recover that

$$
m_2(\Phi_t) \leq e^{t(d+2)}(\beta(1-e^{-2t})+e^{-2t})^{d+2}\left(m_2(\Phi_0) + 2C\sqrt{-2\log(\mu(0)) + d\log\left(\frac{\beta}{2\pi}\right) + d}\right).
$$

Using the expression of $C$, we recover eventually that

$$
m_2(\Phi_t) \leq \frac{2e^{t(d+2)}Z_V^2}{Z_{2V}}(\beta(1-e^{-2t})+e^{-2t})^{d+2}\left[m_2(\mu) + d(\beta+1) - 4\log(\mu(0)) + 2d\log\left(\frac{\beta}{2\pi}\right)\right].
$$

$\square$

## F  Proof of Theorem 9

*Proof.* As Proposition 6 shows, the average error of the estimator can be upper-bounded as

$$
\mathbb{E}[\|\hat{s}_{t,n}(z) - \nabla \log(p_t^V)(z)\|^2] \lesssim \frac{d\log(n)e^{td}Z_{2V}p_t^{2V}(z)}{na(Z_V)^2(p_t^V(z))^2}\left[\frac{\|z\|^2}{(1-e^{-2t})^4} + \frac{b+d}{(1-e^{-2t})^3}\right].
$$

Hence, the average integrated error reads

$$\mathbb{E}[\int \|\hat{s}_{t,n}(z) - \nabla \log(p_t^V)(z)\|^2 \mathrm{d}p_t(z)]$$
$$\lesssim \frac{d\beta^{d+2} \log(n) e^{2t(d+1)}}{na(1 - e^{-2t})^4} \left[ \frac{Z_{2V} m_2(\Phi_t) e^{-t(d+2)}}{(Z_V)^2} + \frac{Z_{2V}(b+d) m_0(\Phi_t) e^{-t(d+2)}}{(Z_V)^2} \right]. \quad (30)$$

We then apply the upper-bounds in Lemma 7 and in Lemma 8 and we recover

$$\mathbb{E}[\int \|\hat{s}_{t,n}(z) - \nabla \log(p_t^V)(z)\|^2 \mathrm{d}p_t(z)] \lesssim \frac{d\beta^{d+3} \log(n) e^{2t(d+1)}(b+d)}{na^2(1 - e^{-2t})^4}$$

We thus set $n$ as $n = d \max(\epsilon^{-2(d+1)+1}, \epsilon^{-5}) = \epsilon^{-2(d+1)+1}$ and we eventually get for $\epsilon \leq t \leq \log(1/\epsilon)$ that

$$\mathbb{E}[\int \|\hat{s}_{t,n}(z) - \nabla \log(p_t^V)(z)\|^2 \mathrm{d}p_t(z)] \lesssim \frac{\epsilon \beta^{d+3}(b+d)}{a^2}.$$

Hence, plugging again the bounds of Lemma 8 in Theorem 3, we recover the desired result. □

