# OpenReview forum: "Sampling from multi-modal distributions with polynomial query complexity in fixed dimension via reverse diffusion"
_NeurIPS.cc/2025/Conference — NeurIPS 2025 poster_

### Official Review · Reviewer_zc68 · 2025-06-18

**Clarity:** 1
**Significance:** 3
**Originality:** 2
**Rating:** 4
**Confidence:** 3

**Summary:**

This paper investigates the problem of sampling from a multi-modal target distribution. Specifically, utilizing the diffusion framework, this paper investigates the problem of estimating the diffusion scores when only the potential of the target is known. The paper provides theoretical score-estimation guarantees for those potentials that satisfy certain smoothness and dissipativity assumptions. Building upon the previous result in (Conforti et al., 2025), this results in an end-to-end sampling guarantee.

**Questions:**

Some questions are in the "Weaknesses" section. Other than those, I have the following questions.

1. In the paper the score estimator is frequently called "our estimator" (even listed as the third contribution). However, the paper also claims that it has been considered in (Saremi et al., 2024). My question is: Is there any algorithmic modification from (Saremi et al., 2024)? If not, my concern is that the third contribution might sound as overclaimed.

2. Is there some other distributional example (which is hard to sample when only the potential is given) that satisfies dissipativity beyond Gaussian mixtures?

3. How are the obtained results related to the sample complexity of estimating the vanilla score-matching objective, in particular for Gaussian mixtures?

I am happy to raise my score if these questions are properly addressed.

**Ethical Concerns:**

["NO or VERY MINOR ethics concerns only"]

**Final Justification:**

The theory part is quite strong. I did not score it higher because I feel that the 2d illustration is still simple, and the dependence on dimension is important (which is not polynomial in the paper).

**Limitations:**

This is a theoretical paper. There is hardly any negative societal impact.

**Paper Formatting Concerns:**

No formatting concerns.

**Quality:**

2

**Strengths And Weaknesses:**

Strengths:
1. Combines diffusion-sampling and unnormalized targets
2. New theoretical guarantees that apply to different target distributions for which the potential is known
3. Polynomial complexity except the dimension

Weaknesses:
1. The contribution on Gaussian mixture seems to be confusing. In particular, it is claimed that "no sampling guarantees were yet derived for general Gaussian mixtures" (line 199). However, this is only true when restricted to the case where the potential function rather than the samples are given. Indeed, there have been many recent attempts to characterize estimation guarantees for Gaussian mixtures under this vanilla objective (e.g., [1]), but none of these is included as related works.

2. Continuing the previous point, the Gaussian mixtures seem to be a special focus both in the motivation and in presenting the results. It is not clear why Gaussian mixtures are important for unnormalized models, as once we know the potential, it is easy to obtain samples and perform vanilla score matching.

3. The presentation of the paper is quite confusing. "Our algorithm" is mentioned in Theorem 1 even before it is proposed. This quite hinders readability.

[1] Sitan Chen, Vasilis Kontonis, Kulin Shah, Learning general Gaussian mixtures with efficient score matching.

---

> ### Author Rebuttal · Authors · 2025-07-30
>
> **Related work on Gaussian Mixtures** -- As noted by the reviewer, the authors of [1] assume that samples of the mixture are available, whereas we assume query access to the potential $V$. While these are two very different settings (Generative modeling vs classical sampling), we will include [1] as a reference in our paragraph on Generative Modeling.
>
> **Sampling from a Gaussian Mixture is easy when $V$ is known** -- We would like to clarify that we do not assume that $V$ is known analytically, only that we can evaluate $V(x)$ at any given point $x \in \mathbb{R}^d$. In particular, even if $V$ is the potential of a Gaussian Mixture, we do not know the corresponding weights, the means and the covariance matrices.
>
> **Result presented before algorithm** -- We followed a standard way of presenting results in TCS (theoretical computer science) [1, 5], where the emphasis is put on the consequences (in this case, polynomial complexity for sampling from a multimodal distribution) rather than on the algorithm itself, which is just a means to an end. However, we fully recognize that clarity is important and when introducing a concept early, we will explicitly state in which later section of the paper it will be elaborated.
>
> **Comparison with Saremi et al., 2024** -- Even though we use the same estimator of the score, note that we do not use it within the same pipeline: in Saremi et al., 2024 the scores are used in a chain of MCMC at a single noise level. In our work, we use this estimator within the reverse diffusion framework, with multiple noise levels. Hence, our overall algorithms differ. Furthermore, note that unlike Saremi et al., 2024, we provide a non-asymptotic error for this estimator (Proposition 4) and we fully control the overall error of our algorithm which is, in our opinion, the main contribution of our work.
>
> **Third contribution is an overclaim** -- We are unsure of the link between the SNIS estimator of the score and our "third contribution": in our introduction, we stated our third contribution as the fact that our algorithm does not require prior knowledge on the target to provably work. Is this what you had in mind?
>
> **Dissipativity beyond Gaussian Mixtures** -- First, we emphasize that our work being the first one to handle Gaussian Mixtures, this seemingly toy case is not an easy one. Furthermore, dissipativity is a quite broad setting: any distribution that is strongly log-concave outside a compact is dissipative and so are all distributions of the form $p(x) \propto e^{- \| x \|^m}$ with $m \geq 2$ (and by stability of dissipativity by mixtures, so are all mixtures of the two previous kinds).
>
> **Comparison with vanilla score matching** -- The asymptotic sample complexity of vanilla score matching scales with the square of the Poincaré constant of the target density (Theorem 2 in [3]). For multi-modal distributions such as Gaussian mixtures, this constant increases exponentially with the between-mode distance (Section 4.1 in [4]). In contrast, our result shows polynomial scaling with respect to the between-mode distance (see our Section 3), which we believe is a key theoretical strength of our work.
>
> We hope we addressed all of your questions and hope you will consider revising your score. Please let us know if any further clarifications are needed.
>
> [1] Chen et al., *Learning general Gaussian mixtures with efficient score matching*, Arxiv 2024
>
> [2] Saremi et al. *Chain of log-concave markov chains*, ICLR 2024
>
> [3] Koehler et al., *Statistical Efficiency of Score Matching: The View from Isoperimetry*, ICLR 2023
>
> [4] Schitling, *Poincare and Log–Sobolev Inequalities for Mixtures*, Entropy 2019
>
> [5] Jason Altschuler and Sinho Chewi, *Faster high-accuracy log-concave sampling via algorithmic warm starts*, Journal of the ACM 2024.

---

> > ### Comment · Reviewer_zc68 · 2025-08-04
> >
> > The authors have cleared up my concerns. I will raise my score.

---

### Official Review · Reviewer_29WY · 2025-06-21

**Clarity:** 3
**Significance:** 2
**Originality:** 2
**Rating:** 4
**Confidence:** 3

**Summary:**

This paper proposes an approach to sampling from unnormalised density functions.
In a nutshell, the sampler starts from a Gaussian initialisation, and denoises along the time-reverse SDE.
To estimate the score, the author proposed to use self-normalised importance sampler using a Gaussian as the proposal.
It shows that this approach has polynomial time complexity in target parameters and can address some limitations that previous works have.

**Questions:**

1. Regarding Fig 1. How do you initialise the Langevin for RDMC? How many steps you use? It is very surprising to me that RDMC failed at this simple 2D problem. I have played around with similar ideas and algorithms before, but they work very well for simple GMMs in a reasonably high dimension.

2. Regarding mode weight.
According to my experience and also discussed in [1], the RDMC-style algorithm (no matter using Langevin or SNIS) has a problem in that it cannot estimate the mode weights accurately. This can be corrected by adding an MH corrector, as done by DiGS [1]. Can the theory reflect this?



[4] Grenioux, Louis, Maxence Noble, and Marylou Gabrié. "Improving the evaluation of samplers on multi-modal targets." FPI workshop @ ICLR 2025.

**Ethical Concerns:**

["NO or VERY MINOR ethics concerns only"]

**Final Justification:**

My concern has been addressed.

- Why not higher:
I feel the proposed algorithm is not strong enough on its own as a simple combination of SNIS and reverse diffusion.

- Why not lower:
My concern has been addressed and I agree with the author the contribution of the theory is significant on its own.

**Limitations:**

Please see Weakness.

**Quality:**

2

**Strengths And Weaknesses:**

### Strength

1. The motivation is clear. This paper found limitations in previous works, and addresses them by SNIS.

2. The theory is solid. I did not check the correctness of all of the proof, but it looks correct to me.

### Weakness

1. **The proposed algorithm is a simple combination of previous works**. Using SNIS to estimate the score is not novel, leveraging the reverse-diffusion with estimated score also exists. In my opinion, the main contribution of this paper is **proving the theorical benefit of using SNIS+DSI for the score**, instead of **proposing a new sampler**.

2. **There are some missing references**. For example, DiGS [1] who use the same Gaussian proposal but with MH instead of SNIS, and  they are not estimating the score, but to re-localise the particles.
Also, while the author cited iDEM, it does not highlight the similarity of the proposed estimator to iDEM's. In short, one can calculate the score by Denoising Score Identity or Target Score Identity, with any estimator, including SNIS, Langevin, etc. Then:

DSI + SNIS -> this paper;

DSI + Langevin -> RDMC;

TSI + SNIS -> iDEM;

TSI with DSI + SNIS -> PDDS [2];

TSI with DSI + Langevin -> DiKL [3].

While most of these papers do not analyse the complexity, they should be discussed as they use very similar estimators.


3. **The experiments are quite toy.** I must first clarify that my primary experience and focus are on the practical side, which may not be directly relevant to this paper's main contribution. The weakness in experiments might be okay for a theoretical paper. But I still have a few concerns.

 - This paper only verifies the results on 2D. It is surprising to me to see RDMC fail on this simple 2D problem. I have a concern that this comparison is not fair. Please see my questions for more on this.

 - I feel it is helpful to change the target parameter (It's fine to do it even in 2D!) and see how the KL / TVD / or any metrics scale with this parameter, given a fixed number of energy evaluations. Or the eval number w.r.t the parameters to achieve a certain KL/TVD/any metrics.
This is much more convincing than just visualising one case.


[1] Chen, Wenlin, et al. "Diffusive gibbs sampling." ICML 2024.

[2] Phillips, Angus, et al. "Particle Denoising Diffusion Sampler." ICML 2024.

[3] He, Jiajun, et al. "Training Neural Samplers with Reverse Diffusive KL Divergence." AISTATS 2025.

---

> ### Author Rebuttal · Authors · 2025-07-30
>
> **Simple combination of previous works** --- Indeed, our algorithm simply combines reverse diffusion + a self normalized estimator of the score. However, that combination having never been explored before, that makes it a new algorithm in its own right. Furthermore, we must insist that our contribution is not the algorithm in itself, which is not very hard to come up with, but rather to study its error and most importantly quantitatively prove that, in terms of query complexity, it is state-of-the-art compared to previous works as it is robust to multi-modality.
>
> **Additional references** --- we have read through the additional references and agree that they are relevant. We appreciate the reviewer's clear summary and DSI / TSI classification. We will add these references and discuss them, especially as they use estimators of the scores which is a central part of our analysis.
>
> **Toy experiment** --- The Figure 1. should not be seen as an experiment but rather as a visual illustration of the performance of our algorithm in comparison with previous works. We highlight the fact that related works have been published at this venue and similar venues without experimental results [1,2,3,4], as the primary contribution is theoretical. That being said, we ran an experiment that was suggested by the reviewer. The distribution to sample is a 2D mixture of three Gaussians with equal weights $1/3$, equidistant modes located at a distance $R$ from the origin, and different covariances $(\mathrm{Id}, \mathrm{Id}/2, \mathrm{Id}/4)$. We allowed each three algorithms $10^6$ queries to the potential $V$ or to its gradient: we performed $10^6$ iterations for ULA, we used $100$ steps of reverse diffusion for our algorithm and RDMC. We used $100$ inner steps of ULA for RDMC that was initialized with a standard gaussian. We used $100$ particles for score estimation for RDMC and $10000$ for our algorithm. Finally, we generated $500$ samples for each three algorithms that we compared in Wasserstein distance to $500$ samples generated from ground truth. As expected, as the between-mode distance grows with $R$, our algorithm yields the lowest error.
>
> |   Between-mode distance R    |  Error of ULA  | Error of RDMC  | Error of our algorithm |
> |:------:|:-----:|:-----:|:-------------:|
> |  1.00  | 0.26  | 1.04  |     0.35      |
> |  3.11  | 0.42  | 3.17  |     0.64      |
> |  5.22  | 4.89  | 6.39  |     0.97      |
> |  7.33  | 8.69  | 9.70  |     1.26      |
> |  9.44  | 12.36 | 12.92 |     2.25      |
> | 11.56  | 16.09 | 16.68 |     3.63      |
> | 13.67  | 18.59 | 19.13 |     5.57      |
> | 15.78  | 21.58 | 22.08 |     3.36      |
> | 17.89  | 23.86 | 24.36 |     3.29      |
> | 20.00  | 28.19 | 28.63 |     4.74      |
>
> **Doubts on poor performance of RDMC on 2D illustration** -- It is not surprising that RDMC fails on this 2D experiment as the dimension *is not* the relevant parameter for RDMC, but rather the isoperimetric constants (Log-Sobolev and/or Poincaré) of the target. Indeed, as we explicitly state in Section 3, RDMC uses ULA to sample from $q_{t, z}(x) \propto e^{-V(x)} e^{- \| x e^{-t} - z \|^2/(2 (1-e^{-2t}))}$ which is almost as hard as sampling from the target $e^{-V}$ as $t$ grows. In particular, in our example, since the target is highly multi-modal, we expect this inner step to fail when $t = O(1)$ and as a consequence, the intermediate scores $\nabla \log(p_t)$ will be poorly estimated.
>
> **Initialization of Langevin for RDMC** --- For RDMC, the inner Langevin step is initialized with a standard gaussian $\mathcal{N}(0, I_2)$. While some improvements are possible [5, 6], they may require unknown parameters of the target distribution (e.g. its smoothness) and the task of sampling from a multi-modal distribution with an inner ULA step remains fundamentally difficult.
>
> **Why our illustration is in 2D** --- Our theory shows that time-reversed diffusion, when combined with a suitable SNIS score estimator, achieves polynomial query complexity *only in low dimensions*. This motivates our choice of a 2D illustration. We emphasize that the target distribution is not a "simple 2D problem": in the absence of prior knowledge of the mode locations, sampling is highly nontrivial. In fact, even in 1D, sampling from this unnormalized density using ULA, for instance, typically incurs exponential query complexity in its parameters (e.g. smoothness), whereas our method retains polynomial complexity.
>
> **Ability to recover mode weights** ---  as is typical in theoretical work, our guarantees are stated in terms of distributional closeness in KL divergence. Explicitly assessing the recovery of mode weights would require a different statistical divergence that is specifically sensitive to mode weights. This is unfortunately beyond the scope of our analysis as well as most of the theoretical literature, to the best of our knowledge. Nevertheless, this is an interesting question for future research.
>
> **Positioning our contribution** --- given their success, time-reversed diffusions have recently gained traction for sampling from unnormalized densities. The reviewer lists many algorithms that are "simple combinations of previous works" yet are still noteworthy. Our method fits this category but stands out by offering rigorous, non-asymptotic theoretical guarantees on query complexity, especially in the presence of score estimation error that we precisely quantify. Such theoretical guarantees are scarce and we believe our results are therefore a meaningful and timely contribution to the field. We hope our responses have addressed the reviewer’s concerns and that they will consider revising their score.
>
> [1] Chen et al., *Sampling is as easy as learning the score: theory for diffusion models with minimal data assumptions*, ICLR 2023.
>
> [2] Lee et al., *Convergence for score-based generative modeling with polynomial complexity*, NeurIPS 2022.
>
> [3] Conforti et al., *KL Convergence Guarantees for Score Diffusion Models under Minimal Data Assumptions*, SIAM 2025.
>
> [4] Chen et al., *Learning general Gaussian mixtures with efficient score matching*, Arxiv 2024
>
> [5] Grenioux et al., *Stochastic Localization via Iterative Posterior Sampling*, ICML 2024
>
> [6] Huang et al., *Faster Sampling without Isoperimetry via Diffusion-based Monte Carlo*, COLT 2024

---

> > ### Comment · Reviewer_29WY · 2025-07-31
> >
> > Thank you for your rebuttal. Your reply addressed most of my concern. I just have one further question:
> >
> > > It is not surprising that RDMC fails on this 2D experiment as the dimension is not the relevant parameter for RDMC, but rather the isoperimetric constants (Log-Sobolev and/or Poincaré) of the target. Indeed, as we explicitly state in Section 3, RDMC uses ULA to sample from $q_{t, z}(x) \propto e^{-V(x)} e^{- | x e^{-t} - z |^2/(2 (1-e^{-2t}))}$ which is almost as hard as sampling from the target $e^{-V}$ as $t$ grows. In particular, in our example, since the target is highly multi-modal, we expect this inner step to fail when $t = O(1)$ and as a consequence, the intermediate scores $\nabla \log(p_t)$ will be poorly estimated.
> >
> >
> > I agree that RDMC uses ULA to sample from $q_{t, z}(x) \propto e^{-V(x)} e^{- | x e^{-t} - z |^2/(2 (1-e^{-2t}))}$, which could lead to significant mode collpase. However, if you randomly initialise this ULA it should work okayish?

---

> > > ### Author Response · Authors · 2025-08-01
> > >
> > > Thank you for your prompt reply, we appreciate.
> > > Indeed, if ULA is initialized with a distribution whose support is "large enough", we expect the target modes to be covered. However:
> > > - the location and the size of this support has no reason to be known a priori, especially with the addition of the $z$ term in $e^{-V(x) - \| x e^{-t} - z \|^2/(2(1-e^{-2t}))}$ which may induce an unknown shift in the location of the modes w.r.t. the initial target $e^{-V}$. This lack of prior information motivated us for the choice a standard gaussian
> > > - even if the support fits, in the case where modes do not have the same weights and/or covariance, ULA will still struggle to converge.

---

> > > > ### Comment · Reviewer_29WY · 2025-08-01
> > > >
> > > > Thank you for your clarification. I agree, but i think it might be better to note this in the title or somewhere about the choice. I will increase my rating

---

> > > > > ### Author Response · Authors · 2025-08-01
> > > > >
> > > > > Thank you for reconsidering your rating. We shall discuss more in depth the choice of the initialization of ULA in the main text for the final version of the paper.

---

### Official Review · Reviewer_uDVT · 2025-07-02

**Clarity:** 2
**Significance:** 3
**Originality:** 2
**Rating:** 4
**Confidence:** 2

**Summary:**

This paper proposes a novel sampling algorithm specifically designed to operate efficiently in scenarios involving multi-modal distributions, which are typically challenging for standard sampling methods. In addition to the algorithmic contribution, the paper provides rigorous theoretical analysis, offering formal guarantees on the performance and convergence of the proposed method.

**Questions:**

Besides the questions mentioned in **Weakness**, there are also several issues.

1. Are there some extension methods which can remove the constrain on dimension?
2. And also, are there some numerical experiments or real world applications to verify the performance of the proposed algorithm?

**Ethical Concerns:**

["NO or VERY MINOR ethics concerns only"]

**Final Justification:**

They addressed my questions, especially on both empirical validation and data dimension constrain, and I have no further questions. So I prefer to keep the positive score 4.

**Limitations:**

Yes.

**Paper Formatting Concerns:**

There is no such issue.

**Quality:**

3

**Strengths And Weaknesses:**

**Strength**
1. The paper provides a clear motivation and presents a well-articulated problem statement that effectively frames the research contribution.
2. The theoretical analysis is thorough, with detailed interpretations and carefully explained assumptions that support the proposed methodology.
3. The proof sketches are clearly written, making the key steps of the theoretical development easy to follow.

**Weakness**
1. The information presented in Figure 1 is somewhat confusing. It would be helpful if the authors could provide additional explanation. Specifically, the meaning of the "dark regime" and "bright regime" should be clarified to enhance the reader's understanding.
2. The experimental validation is lacked. It would strengthen the paper to include experiments that directly verify the theoretical results and further demonstrate the practical applicability of the proposed method.
3. The organization of the paper could be improved. Currently, the introduction presents complex notations that are not fully explained until later, which makes it difficult to follow the story. It may be beneficial to provide some preliminary definitions, notations, or a notation table before Section 3, so that readers can better grasp the main theorem and the subsequent technical content.

---

> ### Author Rebuttal · Authors · 2025-07-30
>
> We thank the reviewer for his careful review and positive comments on our paper.
>
> **Clarifications in the text** --- thank you for picking up on these. We will clarify the caption of Figure 1 with the following explanation. The color scheme indicates the probability density value of the distribution we want to sample from (dark is low probability density, bright is high probability). The blue dots are the samples produced by our algorithm. We will also regroup notations in a paragraph of table as suggested by the reviewer.
>
> **Experimental validation** --- our paper aims to provide strong theoretical guarantees for sampling using time-reversed diffusions. Related works have been published at this venue and similar venues without experimental results [1,2,3,4], as the primary contribution is theoretical. Figure 1 is intended as a visual aid to support the theory. That being said, we ran an experiment that was suggested by reviewer 29WY. The distribution to sample is a 2D mixture of three Gaussians with equal weights $1/3$, equidistant modes  located at a distance $R$ from the origin, and different covariances $(\mathrm{Id}, \mathrm{Id}/2, \mathrm{Id}/4)$. We compare convergence in Wasserstein distance between ULA, RDMC, and our algorithm. As expected,
> as the between-mode distance grows with $R$, our algorithm yields much lower error than ULA and RDMC.
>
> |  Between-mode distance R    |  Error of ULA  | Error of RDMC | Error of our algorithm |
> |:------:|:-----:|:-----:|:-------------:|
> |  1.00  | 0.26  | 1.04  |     0.35      |
> |  3.11  | 0.42  | 3.17  |     0.64      |
> |  5.22  | 4.89  | 6.39  |     0.97      |
> |  7.33  | 8.69  | 9.70  |     1.26      |
> |  9.44  | 12.36 | 12.92 |     2.25      |
> | 11.56  | 16.09 | 16.68 |     3.63      |
> | 13.67  | 18.59 | 19.13 |     5.57      |
> | 15.78  | 21.58 | 22.08 |     3.36      |
> | 17.89  | 23.86 | 24.36 |     3.29      |
> | 20.00  | 28.19 | 28.63 |     4.74      |
>
> **Removing the constraint on the dimension** --- As was stated in our introduction, the exponential dependence in the dimension *cannot* be avoided under our hypotheses because of existing lower-bounds [5,6]. Intuitively, this is because finding the modes of the target distribution would require exploring the entire space, and that search problem grows exponentially with the dimensionality. Our results surprisingly show that, unlike many sampling algorithms, the exponential complexity is *only in the dimension, not in the other parameters* (e.g. smoothness) of the distribution.
>
> [1] Chen et al., *Sampling is as easy as learning the score: theory for diffusion models with minimal data assumptions*, ICLR 2023.
>
> [2] Lee et al., *Convergence for score-based generative modeling with polynomial complexity*, NeurIPS 2022.
>
> [3] Conforti et al., *KL Convergence Guarantees for Score Diffusion Models under Minimal Data Assumptions*, SIAM 2025.
>
> [4] Chen et al., *Learning general Gaussian mixtures with efficient score matching*, Arxiv 2024
>
> [5] Lee et al., *Beyond Log-concavity: Provable Guarantees for Sampling Multi-modal Distributions using Simulated Tempering Langevin Monte Carlo*, NeurIPS 2018
>
> [6] Chak, *On theoretical guarantees and a blessing of dimensionality for nonconvex sampling*, Arxiv 2024

---

> > ### Comment · Reviewer_uDVT · 2025-08-05
> >
> > Thank you to the authors for the response and detailed clarifications. I have no further questions, and I would like to congratulate you on the good work.

---

### Official Review · Reviewer_sQu7 · 2025-07-10

**Clarity:** 4
**Significance:** 3
**Originality:** 3
**Rating:** 5
**Confidence:** 3

**Summary:**

This paper tackles the problem of (approximate) sampling from high-dimensional multi-modal distributions. Multi modality is a common feature of most real-world distributions. In general the problem is known to be computationally intractable, and there are various techniques to speed up sampling that exploit some or the other regularity in the problem description. For multi-modal distributions, polynomial time algorithms are not yet known for situations where there are many unknown modes, and the distribution does not follow extremely stringent smoothness conditions, and if the parameters are not explicitly known.

This paper presents the first algorithm that is able to sample from a class of high-dim. multi-modal distributions that includes Gaussian mixtures, among others. The key idea lies in the use of semi-log-convexity, a relaxation of log-convexity that has some desirable properties. One of which is that a mixture of semi-log-convex densities remains so.

The algorithm uses this for estimating the intermediate score functions required in a reverse diffusion process. Previous methods for score estimation required sampling from a complex distribution, and this paper presents a new way to estimate using ratio of expectations, where both expectations are taken with respect to a simple Gaussian distribution, and are bounded. This combination allows for a polynomial runtime in fixed dimensions. The runtime is exponential in the number of dimensions.

**Questions:**

NA

**Ethical Concerns:**

["NO or VERY MINOR ethics concerns only"]

**Final Justification:**

I think the rating is appropriate for the paper.

**Limitations:**

yes

**Quality:**

4

**Strengths And Weaknesses:**

Strengths:-
1) The paper is generally well written, and the extensive background material is very helpful to understand the paper.

2) The proposed method seems to be novel, and the insights might extend to the broader field of sampling.

Weakness:-
1) Although the paper is mostly of theoretical interest, there are a few illustrative experimental results. I notice that the proposed algorithm, which takes ~1 minute, is significantly faster than the state-of-the-art for the chosen problem. It would have been interesting to see what the algorithm is NOT able to deal with. This might be useful for future research, and also for practitioners when they implement a sampler.

---

> ### Author Rebuttal · Authors · 2025-07-30
>
> **Contributions and future work** --- We appreciate the reviewer’s positive assessment. Time-reversed diffusions have significantly impacted sample generation from multi-modal distributions, and we believe this work offers a timely contribution by providing a strong theoretical foundation for their use to sampling from unnormalized densities. In response to the reviewer’s comment, interesting future directions include developing theoretical guarantees for multi-modal distributions with non-Gaussian tails.

---

### Decision · Program_Chairs · 2025-09-17

**Decision:**

Accept (poster)

**Comment:**

All reviewers praised the novel theoretical contribution of this work, to an important problem relevant to the NeurIPS community.

There is a consensus to accept, but the paper is still borderline; three of four reviewers mark this as only borderline accept. In general, there is concern that the experimental validation is minimal (though this is fine for a theory-oriented paper) and also that the proposed method largely combines existing approaches. Reviewers had mixed opinions regarding the overall clarity of the paper and presentation. There were also some concerns as to the exponential dependency on the dimension, as the polynomial complexity is only for a fixed dimension. However, all reviewers felt on-balance that the theory represents a strong enough contribution on its own for acceptance.